# Space-time first-order correlations of an open Bose Hubbard model with incoherent pump and loss

Martina Zündel[1*], Leonardo Mazza [2,3], Léonie Canet[1,3], Anna Minguzzi[1]

**1** Univ. Grenoble Alpes, CNRS, LPMMC, 38000 Grenoble, France
**2** Université Paris-Saclay, CNRS, LPTMS, 91405, Orsay, France
**3** Institut Universitaire de France, 5 rue Descartes, 75005 Paris
*martina.zuendel@lpmmc.cnrs.fr

## Abstract

We investigate the correlation properties in the steady state of driven-dissipative interacting bosonic systems in the quantum regime, as for example non-linear photonic cavities. Specifically, we consider the Bose-Hubbard model on a periodic chain and with spatially homogeneous one-body loss and pump within the Markovian approximation. The steady state is non-thermal and is formally equivalent to an infinite-temperature state with finite chemical potential set by the dissipative parameters. While there is no effect of interactions on the steady state, we observe a nontrivial behaviour of the space-time two-point correlation function, obtained by exact diagonalisation. In particular, we find that the decay width of the propagator is not only renormalised at increasing interactions, as it is the case of a single non-linear resonator, but also at increasing hopping strength. Furthermore, we numerically predict at large interactions a plateau value of the decay rate which goes beyond perturbative results in the interaction strength. We then compute the full spectral function, finding that it contains both a dispersive free-particle like dispersion at low energy and a doublon branch at energy corresponding to the on-site interactions. We compare with the corresponding calculation for the ground state of a closed quantum system and show that the driven-dissipative nature – determining both the steady state and the dynamical evolution – changes the low-lying part of the spectrum, where noticeably, the dispersion is quadratic instead of linear at small wavevectors. Finally, we compare to a high temperature grand-canonical equilibrium state and show the difference with respect to the open system stemming from the additional degree of freedom of the dissipation that allows one to vary the width of the dispersion lines.

# 1   Introduction

Closed quantum systems under a unitary time evolution have been largely studied in the last two decades, with particular impetus after the experimental reach of the strongly interacting regime both in three-dimensional optical lattices, as well as in lower-dimensional setups [1–4]. One-dimensional interacting quantum systems have attracted much attention since a wealth of theoretical techniques are available, including in particular exact solutions by Bethe Ansatz for homogeneous systems with contact interactions for bosonic or fermionic gases, or mixtures thereof [5–9], and for trapped systems in the limit of infinitely repulsive interactions [10–14]. While lattice fermions are integrable [15], lattice bosons are not, except for the case of the hard-core bosons (see e.g. [16–18]), and the Bose-Hubbard model for two particles [19, 20]. A manifold of attempts to effectively describe non-integrable or 'weakly' non-integrable systems have been put forward, such as the generalised Gibbs ensemble [21], generalised hydrodynamics [22, 23], macroscopic fluctuation theory [24], and these fields are under active investigation.

While all these studies concern closed quantum systems, in several experimental situations quantum systems are subjected to external pump and/or losses, or put in contact with some type of bath. Examples of bosonic open systems are Josephson junction arrays [25], superconducting microwave circuits [26–28], polariton chains [29], excitonic quantum dots coupled to cavity arrays [30–32], atoms or ions in optical cavities [33,34], optomechanical systems [35], driven-dissipative [36] and lossy quantum gases [37–48]. Ultracold bosonic atoms in transport geometries also realise non-equilibrium states (for reviews see e.g. [49,50]). In contrast to the equilibrium case, much less is known for interacting open quantum systems, and several questions arise, ranging from the properties of the steady state and its phase diagram to its coherence, its excitations and its dynamical properties. It is also very interesting to investigate which properties known for unitary systems are still present in open quantum systems, and under which conditions or parameters.

A way to model an open system is to describe it as a subsystem embedded in a larger system acting as a bath and with which it interacts. If one requires the time evolution of the subsystem to be completely-positive and trace-preserving (CPTP map), this evolution has to be of the Lindblad form [51], and described by a Markovian quantum master equation. This evolution follows the composition law for universal dynamical maps which is the quantum analogue to the classical differential Chapman-Kolmogorov equation [52]. Despite its simplicity, the Lindblad master equation suffices to display rich physical behaviour, as it has been demonstrated for the case of the steady state of interacting driven-dissipative bosons [53, 54]. Quadratic Lindbladians, meaning Lindbladians with quadratic Hamiltonian and linear jump operators in the bosonic (fermionic) creation and annihilation operators, are known to be exactly solvable by the method of third quantisation, presented by Prosen [55] and Prosen and Seligman [56]. Within the Keldysh formalism, these systems correspond to a quadratic action. Moreover, instances of quasi-free Lindbladians for fermions and bosons with quadratic hermitian jump operators have been solved [57,58]. Recently, the connection between both formalisms and the phase-space formulation has been pioneered [59]. Non-quadratic Lindbladians require in most cases a numerical solution (see e.g. [60]). We refer to [61] for a comparative analysis of the state-of-art numerical methods, specifically tailored for non-equal time correlation

functions. This method was used to obtain the second-order correlation function $g^{(2)}$ [62].

A particularly important quantity to characterise a quantum system is its two-point correlation function, corresponding to first-order correlations $g^{(1)}$, see e.g. [63]. At equal times, it contains information about the spatial coherence and the presence of (quasi)off-diagonal long-range order, i.e. the existence of Bose-Einstein condensation according to the Penrose Onsager criterion [64]. Its Fourier transform yields the momentum distribution function, giving information on the velocity of the quantum particles in a given quantum state. The full space-time correlation function describes the evolution of the quantum system when removing or adding a particle to it. In space-time, it allows one to study the spread of correlations, and to test the Lieb-Robinson bound [65], and its generalisation to open quantum systems [66]. Its Fourier transform in frequency-wavevector space provides the spectral function, whose poles give the dispersion of single-particle excitations. The spectral function is also needed to describe the transport properties among two reservoirs across a quantum channel within linear response [67].

In this work, we study the two-point correlation function of an interacting bosonic open quantum system described by the Bose-Hubbard model subjected to incoherent drive and losses. At non-zero interactions, similarly to its closed-system counterpart, this model is not integrable. We choose the case of spatially uniform pump and losses, where the non-equilibrium steady-state (NESS) density matrix is exactly known [68], and it is closely related to an infinite-temperature state at fixed chemical potential, independently of the interaction strength. Despite the steady state taking a simple form, we find that the two-point correlation function shows a non-trivial dependence on the interaction strength, displaying in particular typical temporal oscillations with frequency related to the interaction strength as well as an interaction-dependent exponential decay. We compare these results to the exactly known non-interacting limit as well as to the strongly interacting limit, where the dynamics of the different sites decouples and can be described using the exact results for the dissipative Kerr resonator [69]. We then proceed in obtaining the full spectral function, where in particular we find a doublon-like branch at excitation energy $U$ as well as a cosine-like excitation branch in the single-particle-hole sector. The features of the spectral function are a clear signature of the non-equilibrium nature of the quantum system, further supported by a perturbative calculation of the self-energy in the Keldysh formalism, and by the comparison with the closed-quantum system spectral function [70].

The paper is organised as follows. We start by presenting the model in Sec. 2 and characterising the non-equilibrium steady state in Sec. 3. We report our analytical results for the two-point function obtained within the Keldysh formalism in Sec. 4. We then proceed to present our numerical results for the space-time correlations in Sec. 5, where we display the time-resolved correlations and analyse their decay. The spectral function is computed numerically and discussed in Sec. 6. Finally, Sec. 7 presents our concluding remarks.

# 2 Model and observables

## 2.1 Lindblad equation

We consider a lattice of $L$ sites occupied by bosonic particles with on-site repulsive interactions, where each site is homogeneously coupled to a bath allowing for one-body losses and pump. The corresponding unitary evolution is described by the Bose-Hubbard

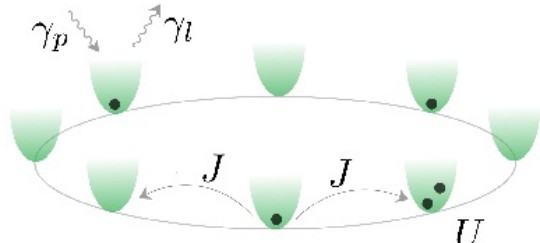

Figure 1: Sketch of the Bose-Hubbard model with periodic boundary conditions, nearest-neighbour hopping amplitude $J$ and on-site interaction strength $U$, coupled to spatially homogeneous Markovian baths with pump rate $\gamma_p$ and loss rate $\gamma_l$. The illustrated couplings apply to all sites.

Hamiltonian:

$$\hat{H} = \sum_{i=1}^{L} \left[ -J\left(b_i^\dagger b_{i+1} + \text{h.c.}\right) + \frac{U}{2} b_i^\dagger b_i^\dagger b_i b_i \right], \tag{1}$$

where $J$ is the hopping amplitude and $U$ the interaction strength. The bosonic operators $b_i$ and $b_i^\dagger$ fulfill the usual commutation relations $[b_i, b_j^\dagger] = \delta_{ij}$, $[b_i, b_j] = 0 = [b_i^\dagger, b_j^\dagger]$, and we take periodic boundary conditions, i.e. we set $L + l \equiv l$.

The full temporal evolution is described by the Lindblad – Gorini-Kossakowski-Sudarsahan equation for the system density matrix $\rho$ [51, 71]:

$$\partial_t \rho = \mathcal{L}[\rho] = -\mathrm{i}[\hat{H}, \rho] + \gamma_l \sum_{i=1}^{L} \left(b_i \rho b_i^\dagger - \frac{1}{2}\{b_i^\dagger b_i, \rho\}\right) + \gamma_p \sum_{i=1}^{L} \left(b_i^\dagger \rho b_i - \frac{1}{2}\{b_i b_i^\dagger, \rho\}\right). \tag{2}$$

The master equation (2) can be obtained from a microscopic model where the system is weakly coupled to a bath, under the assumption that time scales of system and bath are well separated and that the correlations built in the bath do not affect the system (see e.g. [52] for a detailed derivation).

## 2.2 Correlation functions

The two-point correlation function of interest in this work is the retarded Green's function

$$G_{j\ell}^R(t, t') = -\mathrm{i}\Theta(t - t')\left\langle [b_j(t), b_\ell^\dagger(t')] \right\rangle, \tag{3}$$

where $\langle...\rangle$ indicates the trace weighted with the respective density matrix, $\Theta(t)$ is the Heavyside step function, and $b_j(t)$ $(b_j^\dagger(t))$ is the bosonic annihilation (creation) field operator in the Heisenberg picture, time-evolved according to Lindbladian dynamics[1] (in the open case), or Hamiltonian one (in the closed case).

In order to obtain the retarded Green's function for the non-equilibrium steady state (NESS) of the driven-dissipative system, we evaluate unequal time correlation functions, such as

$$\left\langle b_j^\dagger(t) b_0 \right\rangle_{\text{NESS}}^{(\mathcal{L})} \equiv \text{Tr}\left\{ b_j^\dagger e^{\mathcal{L}t}\left[ b_0 \rho_{\text{NESS}} \right] \right\}. \tag{4}$$

---

[1]In the case of open quantum systems, a way of defining the time evolution of an operator $A$ is given by the adjoint superoperator $\bar{\mathcal{L}}$, according to $A(t) = e^{\bar{\mathcal{L}}t}[A]$ where $\bar{\mathcal{L}}[A] = \mathrm{i}[H, A] + \gamma_l \sum_{i=1}^{L} \left(b_i^\dagger A b_i - \frac{1}{2}\{b_i^\dagger b_i, A\}\right) + \gamma_p \sum_{i=1}^{L} \left(b_i A b_i^\dagger - \frac{1}{2}\{b_i b_i^\dagger, A\}\right)$. Notice that $\text{Tr}\{A(t)\rho(0)\} = \text{Tr}\{A\rho(t)\}$.

It will be useful to compare it with the correlation function obtained following a unitary evolution starting from the same NESS density matrix:

$$\left\langle b_j^\dagger(t)b_0\right\rangle_{\mathrm{NESS}}^{(H)} \equiv \mathrm{Tr}\left\{\mathrm{e}^{\mathrm{i}Ht}\, b_j^\dagger\mathrm{e}^{-\mathrm{i}Ht}\, b_0\rho_{\mathrm{NESS}}\right\} . \tag{5}$$

For a closed system the corresponding correlation function evaluated on the ground state $|\Psi_0\rangle$, which reads

$$\left\langle b_j^\dagger(t)b_0\right\rangle_{\mathrm{GS}} \equiv \langle\Psi_0|\mathrm{e}^{\mathrm{i}Ht}\, b_j^\dagger\mathrm{e}^{-\mathrm{i}Ht}\, b_0|\Psi_0\rangle , \tag{6}$$

and evaluated on a finite temperature equilibrium ensemble, which takes the form

$$\left\langle b_j^\dagger(t)b_0\right\rangle_{\mathrm{equi}} \equiv \mathrm{Tr}\left\{\mathrm{e}^{\mathrm{i}Ht}\, b_j^\dagger\mathrm{e}^{-\mathrm{i}Ht}\, b_0\rho_{\mathrm{equi}}\right\} , \tag{7}$$

where $\rho_{\mathrm{equi}}$ is the canonical $\rho_C(\beta, N) = \mathrm{e}^{-\beta\hat{H}}/\mathrm{Tr}[\mathrm{e}^{-\beta\hat{H}}]$ or grand-canonical density matrix $\rho_{GC}(\beta,\mu) = \mathrm{e}^{-\beta(\hat{H}-\mu\hat{N})}/\mathrm{Tr}[\mathrm{e}^{-\beta(\hat{H}-\mu\hat{N})}]$ with $\beta^{-1} = k_B T$.

We finally consider the spectral function, whose poles provide the excitation branches of the system, and which is defined as

$$A(\omega, k) = -\frac{1}{\pi}\mathrm{Im}\, G^R(\omega, k) \tag{8}$$

for a homogeneous system.

# 3 Exact properties of the non-equilibrium steady state

## 3.1 Steady-state density matrix

It has been shown in Ref. [68] that the Ansatz

$$\rho_{\mathrm{NESS}} = \frac{1}{\mathcal{N}}\sum_{N=0}^{\infty} z^N\mathbb{1}_N , \qquad z = \frac{\gamma_p}{\gamma_l} , \tag{9}$$

is a NESS of the Lindblad equation (2) for $z < 1$. Notice that this state corresponds to an infinite-temperature grand-canonical partition function with fugacity $z$. The normalisation $\mathcal{N}$ in Eq. (9) is given by

$$\mathcal{N} = \sum_{N=0}^{\infty} D_{L,N}^B z^N , \qquad D_{L,N}^B = \mathrm{Tr}\,\mathbb{1}_N = \left(\!\!\binom{L}{N}\!\!\right) \equiv \binom{L-1+N}{N} , \tag{10}$$

where we used the fact that the dimension of the Hilbert space for $N$ bosons in a system of length $L$ is given by the number of $L$-tuples of total length $N$. This is mathematically equivalent to the 'star and bar' problem of identifying in how many ways one can place $L-1$ bars (separating sites) between $N$ stars (occupied sites). The normalisation can be further simplified to $\mathcal{N} = (1-z)^{-L}$. A generalised formula for the normalisation in a truncated Hilbert space can be found in Appendix B.

We remark that, in this idealised model, there is no high-energy cutoff on the pump intensity. We refer to Ref. [68] for a model with frequency dependent effective jump operators. The infinite dimensional Hilbert space of the model as written above renders the Hamiltonian unbounded. We point out that in the numerical calculations, we have to introduce a cutoff for the local occupation, which gives an upper bound for the Hamiltonian and ensures that the time evolution is a CPTP map as well as the existence of a NESS [52]. We have generalised the solution (9) for $\rho_{\mathrm{NESS}}$ to the case of driven-dissipative free fermions and hard-core bosons on the lattice. The results are given in Appendix C and Appendix D respectively.

## 3.2   Equal-time correlations in the steady state

The simple form of the steady state solution (9) allows one to exactly calculate the equal-time correlation function of the bosonic field operators in the NESS. We here present the solution for the one-body density matrix $\langle b_i^\dagger b_j \rangle$ and for the variance of the local particle occupation number $\mathrm{VAR}(n_i)$, as they give insights in the occupation of the system and fluctuations around it. The higher moments can be easily obtained following the same method. The equal-time correlation function in the NESS is given by

$$
\langle b_i^\dagger b_j \rangle_{\mathrm{NESS}} = \mathrm{Tr}\{b_i^\dagger b_j \rho_{\mathrm{NESS}}\}
$$

$$
= (1-z)^L \sum_{N=0}^{\infty} z^N \sum_{\{m_r\}_{r=1,\dots,L}}^{\prime} \langle \{m_r\}|b_i^\dagger b_j|\{m_r\}\rangle \,, \tag{11}
$$

where we inserted a complete basis in the Fock space $|\{m_r\}\rangle$ for $N$ particles on $L$ sites, with $b_r^\dagger b_r|\{m_r\}\rangle = m_r|\{m_r\}\rangle$, and where $\sum'_{\{m_r\}_{r=1,\dots,L}}$ runs over the sets $\{m_r\}$ such that $\sum_r m_r = N$. We note that, due to the diagonal form of $\rho_{\mathrm{NESS}}$, the matrix elements $\langle \{m_r\}|b_i^\dagger b_j|\{m_r\}\rangle$ are non-vanishing only if $i=j$. After some combinatorial manipulations (see analogous derivation in Appendix A.3), we find that

$$
\langle b_i^\dagger b_j \rangle_{\mathrm{NESS}} = \delta_{ij}\frac{z}{1-z} \,, \tag{12}
$$

where one must assume that $z < 1$, i.e. $\gamma_l > \gamma_p$. The loss rate must be larger than the pump rate, in agreement with the Keldysh analysis of Sec. 4. Note that since $\langle b_i^{(\dagger)}\rangle_{\mathrm{NESS}} = 0$, the correlation function (12) is also equal to the connected one.

    With a similar method, we next calculate the fluctuations of the local particle number density. This quantity is useful in order to assess the effect of the truncation of the local Hilbert space in the numerical calculations. The variance is given by

$$
\mathrm{VAR}(n_i) \coloneqq \langle n_i^2 \rangle - \langle n_i \rangle^2 = \frac{z}{(1-z)^2} \,. \tag{13}
$$

We note that at vanishing pump rate $z \to 0$ the local number fluctuations vanish, while they diverge for $z \to 1$. In the latter regime the system is close to the instability point. The ratio of the number fluctuation $\Delta(n_i) = \sqrt{\mathrm{VAR}(n_i)}$ to the average occupancy $\langle n_i \rangle$ is given by $\frac{1}{\sqrt{z}}$ and decreases for large occupation. This calculation provides a good estimate on the error made by truncating the Hilbert space to $N_s$ states per lattice site. For example, choosing $z = 1/10$ ($z = 1/5$) gives an average occupation of $1/9$ ($1/4$) with the number fluctuation of $\sqrt{10}/9$ ($\sqrt{5}/4$). Hence, for these parameters, a truncation of the Hilbert space to $N_s = 3$ allowing only for the states $\{|0\rangle, |1\rangle, |2\rangle\}$ on each site is justified. This will be confirmed by the numerics in the following.

    For the total particle number $N = \langle \sum_{j=1}^L b_j^\dagger b_j \rangle$ in the system, one can solve the full time evolution analytically. The result reads

$$
\langle N(t) \rangle = (N_0 - N_{\mathrm{NESS}})\,\mathrm{e}^{-(\gamma_l - \gamma_p)t} + N_{\mathrm{NESS}} \,, \tag{14}
$$

where $N_0 = N(t=0)$ and $N_{\mathrm{NESS}} = \frac{z}{1-z}L$ is the total steady-state occupation. The details of the calculation can be found in Appendix A.2. Notice that there exists a NESS if and only if the pump rate is chosen strictly smaller than the loss rate, i.e. $z < 1$, consistently with the previous considerations.

# 4 Schwinger-Keldysh formalism

In this section, we use field-theoretical methods for out-of-equilibrium systems to determine the corrections at small interaction to the self-energy, which yields the corrections to the dispersion and decay. We first rewrite the model in the Schwinger-Keldysh formalism, following the derivation of [54], and define the physical observables such as the response function, dispersion lines and decay width (inverse life time of the excitations). We then calculate in a perturbative expansion at small $U$ the corrections to the self-energy at one-loop order.

## 4.1 Keldysh action and response function

Starting from the Lindblad equation (2), and expanding the system density matrix onto two coherent-state bases for each space-time point identified by the fields $\varphi_j^+(t)$ $(\varphi_j^-(t))$ for upper (lower) time contours, one obtains the time evolution of the density matrix in terms of a path integral. Employing the usual notations, the Keldysh action in the rotated basis $(\varphi_{c,j}(t), \varphi_{q,j}(t)) = \frac{1}{\sqrt{2}}(\varphi_j^+(t) + \varphi_j^-(t), \varphi_j^+(t) - \varphi_j^-(t))$ reads

$$
S = \int dt \sum_j \left[ \varphi_{c,j}^* \, i \left( \partial_t - \kappa_0 \right) \varphi_{q,j} + \varphi_{q,j}^* \, i \left( \partial_t + \kappa_0 \right) \varphi_{c,j} + i\gamma |\varphi_{q,j}|^2 \right.
$$
$$
+ J \left( \varphi_{c,j+1}^* \varphi_{q,j} + \varphi_{q,j+1}^* \varphi_{c,j} + \varphi_{c,j}^* \varphi_{q,j+1} + \varphi_{q,j}^* \varphi_{c,j+1} \right)
$$
$$
\left. - \frac{U}{2} \left( (\varphi_{c,j}^2 + \varphi_{q,j}^2)\varphi_{c,j}^* \varphi_{q,j}^* + (\varphi_{c,j}^{*\,2} + \varphi_{q,j}^{*\,2})\varphi_{c,j}\varphi_{q,j} \right) \right], \qquad (15)
$$

with

$$
\kappa_0 = \frac{\gamma_l - \gamma_p}{2} , \qquad \gamma = \gamma_l + \gamma_p . \qquad (16)
$$

In the same notation, the retarded Green's function (3) reads

$$
G_{j\ell}^R(t, t') = -i \left\langle \varphi_{c,j}(t)\varphi_{q,\ell}^*(t') \right\rangle_c , \qquad (17)
$$

where the $c$ index stands for connected.

The saddle-point approximation in the path integral with the action (15) allows one to study the existence and stability of a condensate. For a static homogeneous field solution, if $\gamma_p < \gamma_l$, the imaginary part of the Keldysh potential has a global minimum for $\varphi_{c,j} = 0$, $\varphi_{q,j} = 0$, which implies that the system is in the symmetric phase without breaking of the U(1) symmetry. Hence, there is no finite order parameter. As we saw before, this is in agreement with the exact NESS solution. Notice that this is at variance with the case where both one- and two-body losses are present, where a U(1) symmetry-broken phase emerges for $\gamma_p > \gamma_l$ and a (quasi-)condensate forms at weak interactions [72].

From the retarded sector of the quadratic action, we obtain the spectral lines

$$
\Omega_\pm(k) = -i\kappa_0 \mp 2J\cos(k) , \qquad (18)
$$

as detailed in Appendix E.2. This approach allows us to identify $\kappa_0$ with the decay rate at weak interactions corresponding to the broadening of the spectral line, as well as the excitation dispersion $\Omega_+$, corresponding to the free-particle dispersion in a lattice. From the Keldysh formalism, we obtain the retarded Green's function

$$
G_0^R(k, \omega) = \frac{1}{\omega + 2J\cos(k) + i\kappa_0} , \qquad (19)
$$

which has poles for $\omega = \Omega_+(k)$. The averaged particle number density number $\langle n_j \rangle$ can also be obtained within the Keldysh formalism, by calculating the Keldysh Green's function at equal times $G_{jj}^K(t,t)$. Details can be found in Appendix E.3. We obtain $\langle n \rangle = \frac{\gamma_p}{\gamma_l - \gamma_p} = \frac{z}{1-z}$, which does not depend on $j$ for a homogeneous system, and which coincides with the exact result (12).

To include the effect of interactions, we perform a perturbative calculation in $U$ in the following section and then turn to numerical simulations to treat arbitrary $U$ in Secs. 5 and 6 below.

## 4.2 One-loop correction

We detail in Appendix E.4 the calculation of the one-loop correction to the self-energy $\Sigma$, i.e. the linear correction term in the coupling $U$ to the inverse bare propagator $G_0^{-1}$. By means of the Dyson equation, the Green's function $G$ is related to the self-energy as

$$G^{-1} = G_0^{-1} - \Sigma \ . \tag{20}$$

We obtain, at one-loop order, the retarded Green's function as

$$G_{1-\text{loop}}^R(k,\omega) = \frac{1}{\omega + 2J\cos(k) - \Sigma_1 + i\kappa_0} \ , \qquad \Sigma_1 = \frac{\gamma U}{2\kappa_0} = U(2\langle n \rangle + 1) \ . \tag{21}$$

This result, derived in Appendix E.4, shows that the self-energy to linear order in the coupling depends on the filling of the bosons $\langle n \rangle$, which is tunable through pump and loss rates. Furthermore, this contribution only alters the position of the dispersion lines, but does not change their width. We can hence predict that, for low interaction strength $U \le J$, the dispersion lines are shifted by

$$\Omega_{\pm,1-\text{loop}} = -i\kappa_0 \mp \left(2J\cos(p) - \frac{\gamma U}{2\kappa_0}\right) \ . \tag{22}$$

In order to obtain the local decay of the retarded Green's function defined in (3) and section 5.3, we compute the inverse Fourier transform of (21). One finds

$$G_{jj}^R(t,0) = \int_{-\infty}^{\infty} \frac{d\omega}{2\pi} e^{-i\omega t} \int_{-\pi}^{\pi} \frac{dp}{2\pi} G_{1-\text{loop}}^R(p,\omega)$$
$$= -i\, J_0(2Jt) e^{-\kappa_0 t - i\frac{\gamma U}{2\kappa_0}t}\, \Theta(t) \ , \tag{23}$$

where $J_0(x)$ is the Bessel function of first kind.

# 5 Correlation function in real time

We report in this section our numerical results for the non-equal time two-point correlation function.

## 5.1 Analysis of the parameter space

The original model (2) depends on four couplings: $\{J, U, \gamma_l, \gamma_p\}$. By rescaling time by the typical loss time $\tau_l = 1/\gamma_l$, we are left with the three independent parameters $\{J/\gamma_l, U/\gamma_l, z\}$, with $z = \gamma_p/\gamma_l$, for which $J \in \mathbb{R}_0^+$, $U \in \mathbb{R}_0^+$ and $z \in [0,1)$ for a NESS to exist. The parameter space and known limiting cases are illustrated in Fig. 2. The plane $z = 0$ corresponds to a purely lossy system with empty steady state. The plane $J = 0$

corresponds to the case of absence of tunneling among the sites – in this case the problem maps to a set of single-site dissipative Kerr resonator which has an exact solution [69]. The plane $U = 0$ corresponds to the non-interacting limit which can be solved exactly using the non-interacting (Gaussian) Keldysh action. In the following, we fix a value for $z$ and explore numerically the two-point correlation function in the interaction/tunnel energy plane.

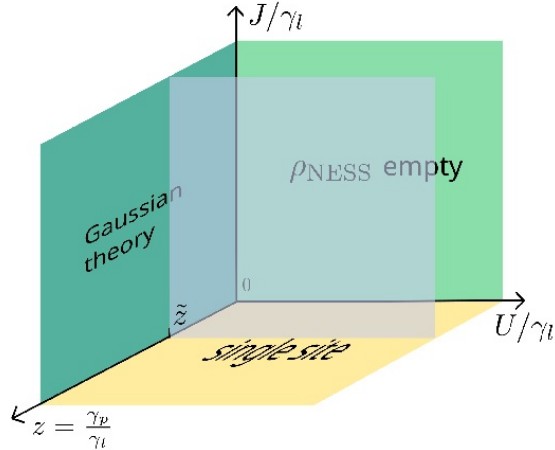

Figure 2: Sketch of the parameter space under investigation: $J/\gamma_l \in \mathbb{R}_0^+$, $U/\gamma_l \in \mathbb{R}_0^+$ and $z \in [0, 1)$; the plane $z = 0$ (light green) describes a purely lossy system with empty steady state; the plane $J = 0$ (yellow) corresponds to a local problem with a known exact solution for the $L$ independent dissipative Kerr resonators; the plane $U = 0$ is the non-interacting limit with also an exact solution. The inserted plane for finite $z$ indicates the numerically investigated phase space.

## 5.2 Time correlations

Using the analytical expression for the NESS density matrix, we have performed numerical calculations using exact diagonalisation (see details in Appendix G.1) to obtain the retarded Green's function in space-time. Fig. 3 displays an example of such a Green's function, which displays the emergence of two main features. First, the Green's function decays exponentially in time, and second, it oscillates at well-defined frequencies. Quite remarkably, both the decay rate and oscillation frequencies are affected by the interactions. We next focus on the properties of the decay rate $\kappa$. The frequencies are discussed in Sec. 6 and Appendix A.4.

## 5.3 Renormalisation of the temporal decay

In order to capture the main features of the non-static properties in the NESS, we determine the inverse life time, i.e. the decay $\kappa$ of the equal-space retarded Green's function. We investigate how the decay $\kappa(U, J)$ changes for fixed ratio of pump-to-loss rate $z$. As a benchmark of our numerical procedure, we recover the known exact results, both in the limit of weak interaction and in the limit of weak hopping.

To extract the decay rate from the numerical data, we assume that the retarded Green's function $G_{00}^R(t) \equiv G_{00}^R(t, 0)$ endows at coincident spatial points a single-pole form corresponding to

$$G_{\text{sp}}^R(t) = -\mathrm{i}\Theta(t)\mathrm{e}^{-\mathrm{i}\Omega t - \kappa t} \; . \tag{24}$$

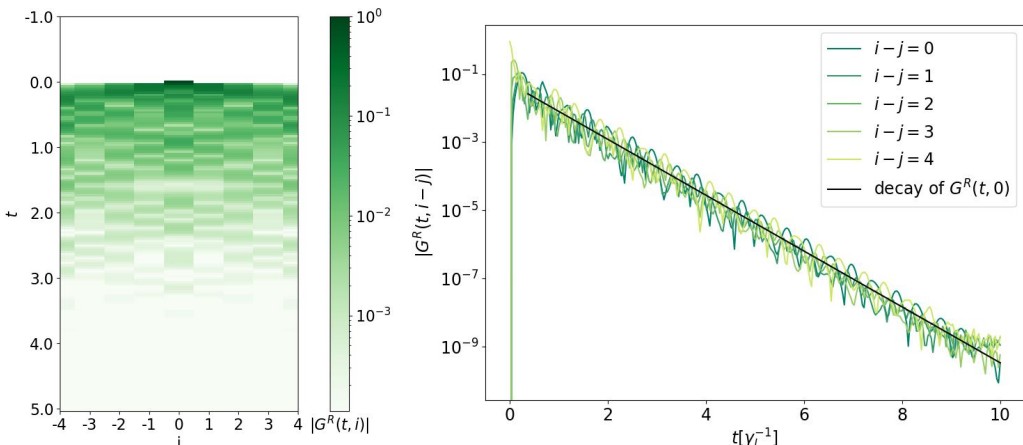

Figure 3: Left panel: space-time evolution of the real part of the retarded Green's function (dimensionless). Right panel: cuts for various spatial differences as indicated on the legend as a function of time (in units of $\tau_l = 1/\gamma_l$) and extracted exponential decay rate (solid black line). All results are obtained by exact diagonalisation with parameters $U = 50$, $J = 10$, $z = 0.2$, $L = 8$, $N_s = 3$.

This leads us to define the decay rate as $\kappa = -\Re\left[\ln i G_{\text{sp}}^R(t)\right]/t$ for $t > 0$. We thus represent the numerical data as $\mathcal{G}(t) = -\Re\left[\ln i G_{00}^R(t)\right]$ and determine the slope of $\mathcal{G}$ to obtain an estimate of the decay rate (see Appendix G.2 for details). The resulting decay rate is depicted in Fig. 4 in the $\{U, J\}$ plane. At weak interactions, i.e. for $U/\gamma_l \ll 1$, we recover the prediction from the free, quadratic theory of Sec. 4, given by Eq. (16). Furthermore, the one-loop calculation demonstrates that the corrections are beyond linear order in the interaction. This prediction is valid for small interaction up to $U/\gamma_l \lesssim 1$ and takes into account a finite hopping $J$. We indeed observe in our numerical result that the behaviour of the decay rate at small $U$ is slower than linear.

At strong interactions, when tunneling is negligible with respect to both interaction energy $U/J \gg 1$ and pump/loss rates $J \ll \gamma_l, \gamma_p$, the decay rate tends to a plateau independent of the interaction strength which was, to the best of our knowledge, unknown for extended systems. For the single-site case, the value of this plateau can be extracted from the exact solution of Ref. [69] for a single dissipative Kerr resonator as

$$\kappa_\infty = \frac{\gamma_l + 3\gamma_p}{2} = \kappa_0 + 2\gamma_p \,, \tag{25}$$

(see Appendix F for details). In between these two limits, we observe a renormalisation of the decay smoothly connecting weak and strong interactions, and weak and large decay rate, similarly to the predictions for the single-site case [69]. We have checked that our numerical simulation for $L = 1$ agrees well with the exact solution all through the crossover from weak to strong interactions. Let us emphasise that the existence of the plateau and its value $\kappa_\infty$ (Fig. 4) generalise the results previously known for a single Kerr resonator to extended systems. This was obtained numerically as it lies beyond the reach of perturbative calculation.

We notice that the decay rate increases both when the interaction strength increases at fixed $J$, as well as when the tunnel amplitude increases at fixed (large) $U$. We understand this latter effect as being due to increased decay possibilities upon allowing for tunneling among particles. For the parameter regime we could access numerically, we find that the value of the decay rate for large tunneling amplitude and large interaction strengths

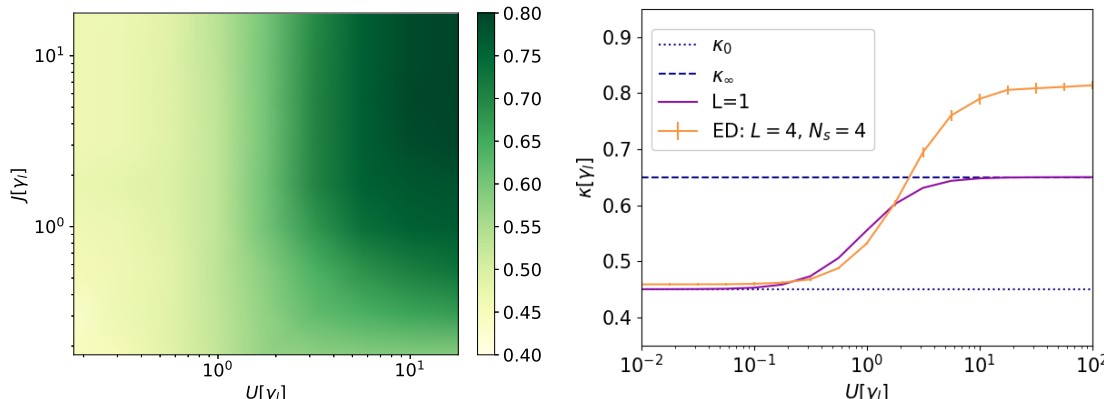

Figure 4: Left panel: decay rate $\kappa$ of the retarded Green's function for $z = 0.1$ in the $\{U, J\}$ plane with $U$, $J$ in units of $\gamma_l$. Right panel: cut of the decay rate as a function of $U/\gamma_l$ at fixed $J/\gamma_l = 10$. The analytical solution for a single site Eq. (85) is also shown, as well as the two limits $\kappa_0 = \kappa(U = 0)$ from Eq. (16) and $\kappa_\infty = \kappa(U \to \infty)$ from Eq. (25). Calculations are done by exact diagonalisation with $L = 4$, $N_s = 4$. 8

depends on the system size (compare to Appendix G.4).

# 6 The spectrum

We now focus on the properties of the single-particle excitations on top of the NESS by analysing the spectral function $A(k, \omega)$ defined in Eq. (8). For interacting 1D bosons in closed systems, this function displays several noticeable features as broad spectra with power-law singularities in correspondence of the Lieb-I and Lieb-II excitation branches [73]. At low energy, the dispersion of the Goldstone branch is linear, confirming the $z = 1$ dynamical critical exponent of the equilibrium model. On the lattice, the spectral function also displays a third excitation branch associated with the change of curvature of the single-particle dispersion [17], as well as a doublon branch at energy close to $U$ [74]. These results are used as a reference to analyse and discuss the spectral function of the open system.

## 6.1 Spectrum of the open system

Our exact diagonalisation results for the spectral function are shown in Fig. 5. At weak interaction, we find the free spectral line with the Lorenzian shape in frequency predicted from the field theory calculation in Sec. 4. Two main features emerge at strong interactions. First, differently from the closed-system case, the low-energy branch is quadratic rather than linear, with an emerging dispersion branch corresponding to the free particle one. The background of excitations behind this main branch is due to the $N > 1$ particle sectors in $\rho_{\text{NESS}}$. They provide additional contributions to the spectral function whose precise shape is difficult to resolve given the system size. Second, we identify an additional excitation branch centered around energy $U$, which is the analogue of the doublon branch of the closed-system case. This branch provides a clear signature of interactions. At first glance this seems astonishing, as $\rho_{\text{NESS}}$ does not contain information on the interaction itself and does not depend explicitly on $U$, in contrast to the equilibrium density matrix $\rho_{GC}$. We provide an estimate of the position of this excitation branch using a strong-coupling approach in Sec. 6.2.

The spectral function here obtained has limited resolution due to the size of the system accessible in numerical diagonalisation, but displays nevertheless emergent features characteristic also of larger system sizes. This is illustrated in Sec. 6.3, where we present the calculation of the spectral function for the closed-system case for two choices of system size, which indeed present similar features.

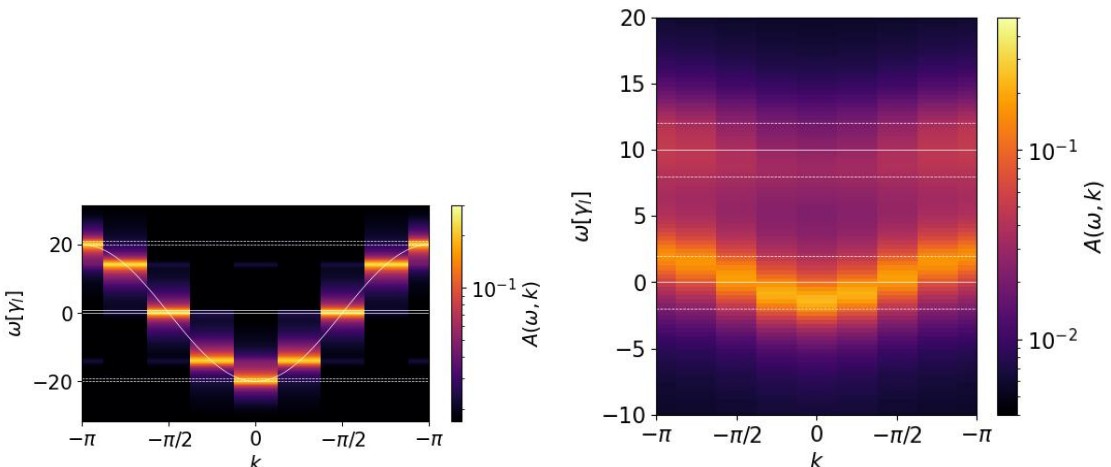

Figure 5: Spectral function $A(\omega, k)$ for the NESS of a driven-dissipative Bose-Hubbard model in the weakly (left panel, $U/\gamma_l = 1$, $J/\gamma_l = 10$) and strongly (right panel $U/\gamma_l = 10$, $J/\gamma_l = 1$) interacting regimes. The other parameters are: $z = 0.2$, $L = 8$, $N_s = 3$. The horizontal white lines are set at $\omega_b := kU$, $k \in \{1, ... N_s - 1\}$ (positions of the branches for closed system and $J = 0$, compare with (29)), and dotted lines at $\omega = \omega_b \pm 2J$.

## 6.2 Position of the excitation spectral lines for small hopping

In order to estimate the position of the excitation branches in the interacting case, we analytically calculate the oscillation frequency of the doublon branch of the NESS under unitary time evolution. As shown in Sec. 6.5, the unitary time evolution well accounts for the position of the peaks of the spectral function.

We use the single site model, i.e. we set $J = 0$. In this case, the Hamiltonian becomes local in position space

$$H = \frac{U}{2} \sum_j n_j(n_j - 1) \ . \tag{26}$$

Upon evaluating the two-point correlation function in the NESS,

$$\left\langle b_j^\dagger(t) b_0 \right\rangle_{\text{NESS}} = (1 - z)^L \sum_N z^N \sum_{\{m\}} \langle \{m\} | e^{iHt} b_j^\dagger e^{-iHt} b_0 | \{m\} \rangle \ , \tag{27}$$

we obtain that the only non-vanishing term is the one with $j = 0$ due to the locality of the strong-coupling Hamiltonian. This term reads

$$\langle m_0 | e^{i\frac{U}{2} n_0(n_0-1)t} b_0^\dagger e^{i\frac{U}{2} n_0(n_0-1)t} b_0 | m_0 \rangle = m_0 e^{iU(m_0-1)t} \ . \tag{28}$$

This shows that there is an oscillation with period depending on the interaction strength $U$. Altogether, we find

$$\left\langle b_j^\dagger(t) b_0 \right\rangle_{\text{NESS}} = \delta_{0j} z(1 - z) \sum_{m=0}^\infty z^m (m + 1) e^{iUmt} \ , \tag{29}$$

with $m + 1$ corresponding to the occupation number of the site $j = 0$. Further details of the calculation can be found in Appendix A.4. We notice from Eq. (29) that the position of the first excitation peak for vanishing $J$ and large $U$ corresponds to the contribution with $m = 1$, i.e. $\omega_d(m = 1) = U$. This is at the origin of the doublon branch in Fig. 5. Furthermore, we observe that the number of dominant peaks in the excitation spectrum at $k = 0$ is exactly $N_s - 1$, as predicted by (29).

## 6.3    Spectral function for the ground state of the Bose-Hubbard model

In order to assess the role of drive and dissipation on the spectral function, we show here the results for the spectral function of a closed Bose-Hubbard system at small filling for the same values of parameters, namely interactions and hopping strength, as well as the same system size. In the closed system, numerical calculations allow one to tackle larger system sizes, and to explore the interplay between interaction and finite-size effects in the spectral function. As shown in Fig. 6, features clearly identified for large system size are also found at smaller size and serve as guideline to interpret our results for the open case.

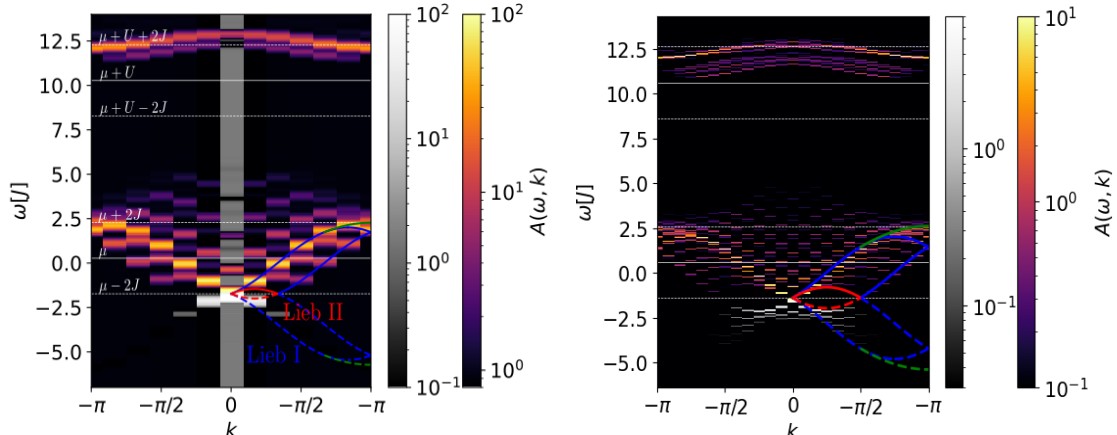

Figure 6: Spectral function $A(k, \omega)$ for the ground state of the Bose-Hubbard model under unitary evolution (negative values in grey). The blue and red lines indicate the Lieb-I and II branches respectively, the green branch is a feature of the lattice model, see text. Left panel: Exact diagonalisation solution without truncation in the local Hilbert space at finite interaction ($L = 12$, $N = 2$ particles, $U/J = 10$), compare with Appendix G.5. Right panel: Larger system size ($L = 24$, $N = 6$, $U/J = 10$, with $N_s = 3$ obtained with DMRG).

Let us comment on the results for the closed system. In the lowest-energy excitation manifold, we clearly identify the Lieb-I and Lieb-II branches, as well as the third branch predicted for the lattice case [17]. The predictions for the dispersion branches in the infinite interaction limit is also shown in Fig. 6. Notice that in this limit all the branches are slightly shifted upwards with respect to the finite-$U$ spectrum, since they are centered at $\omega = \mu - 2J$ and the value of the chemical potential $\mu = \pi^2 J(\frac{N}{L})^2$ at infinite interactions is larger than the one at finite interactions. At frequencies $\omega \simeq \mu + U + 2J$, we also see in the figure a dispersive doublon branch.

We stress that this result is very different from the one obtained by the Lindblad evolution presented in the previous section. In the latter case, the spectral function displays a quadratic behaviour at low momenta, while in the current case the dispersion at low momenta is linear. This is an illustration of the fact that the nature of the low-energy excitations involved is different depending on the state of the system considered. For the

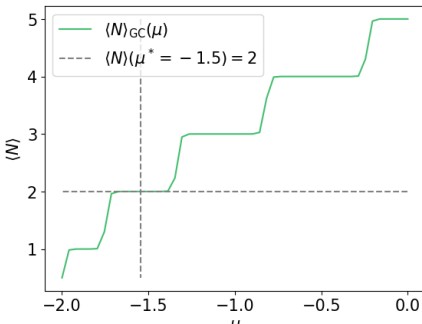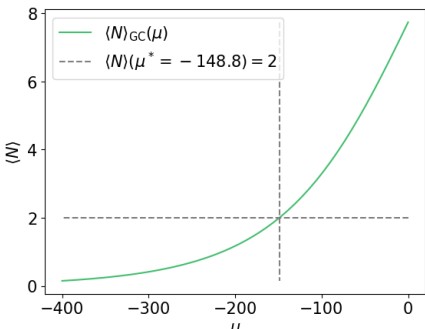

Figure 7: Average filling $\langle N \rangle$ of the system depending on the chemical potential $\mu$ given a grand-canonical distribution (30), with system parameters $U/J = 10$ $L = 8$, $N_s = 3$. Left panel: Low-temperature regime, with $\beta = 100J^{-1}$. The Mott lobes correspond to the plateaus. Right: High-temperature $\beta = 0.01J^{-1}$.

ground state, low-energy excitations are particle-hole excitations on top of an effective Fermi sphere, while for the NESS state, all excitations are possible, and single particle excitations dominate the intensity of the spectral function.

## 6.4 Spectral function of the finite-temperature equilibrium state

In this section, we calculate the spectral function of an equilibrium state at finite temperature, and we highlight the differences with the spectral function of the open quantum system.

### 6.4.1 Equilibrium spectral function

The spectral function is obtained by performing thermal averages (see again Eq. (7)) of the two-point correlations functions, assuming a grand-canonical density matrix

$$\rho_{GC}(\beta,\mu) = \frac{1}{Z}\mathrm{e}^{-\beta(H-\mu N)} \quad , \tag{30}$$

with $Z = \mathrm{Tr}[\mathrm{e}^{-\beta(H-\mu N)}]$. In order to compare the results with the spectral functions obtained in the open system, we choose the same size and average filling of the system in the two cases. We obtain the chemical potential $\mu$ by numerically calculating the average occupation $\langle N \rangle(\mu)$ and inverting the relation, as illustrated in Fig. 7. The spectral function is shown in Fig. 8 for two values of the temperature. In the extremely low-temperature regime (left panel), the spectral function is very close to the ground-state results presented in Fig. 6, as expected. In the extremely high-temperature regime (right panel) the spectral lines are blurred with a broadening which, in the upper branch, reaches the whole bandwidth. We stress that the linewidth of the high-temperature case is different from the case of a NESS, shown in Fig. 5, although corresponding to the same values of interaction strength and filling. This shows that although the NESS formally resembles an infinite-temperature state, the time-dependent observables on top of this state are different. Note, that the system is not at infinite temperature, as out-of-equilibrium there is no temperature. Moreover, there is an additional degree of freedom, which is the pump to loss ratio $z$. The comparison of Fig. 5 (right panel) to Fig. 9 (left panel) shows that, by choosing a different ratio of the coherent to the dispersive parameters –while keeping the filling and the ratio $J/U$ constant– one can tune the linewidth, which is not possible in equilibrium.

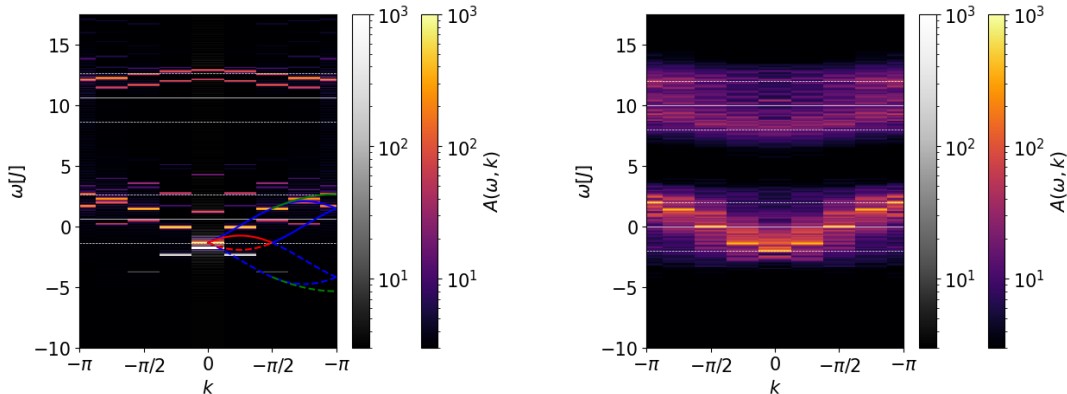

Figure 8: Spectral function within an equilibrium grand-canonical state (30) at strong interactions $U/J = 10$. Left panel: Low-temperature regime $\beta = 100J^{-1}$ (comparison with exact-diagonalisation solution as in Fig. 6). Right: High-temperature regime $\beta = 0.01J^{-1}$. The average particle number is set to $\langle N \rangle = 2$ in order to have the same filling as in the open-system case. The other parameters used are $L = 8$, $N_s = 3$.

A more general argument for showing that the NESS (9) is different from a thermal-equilibrium state is obtained by examining the action (15): the latter reveals an explicit breaking of the time-reversal symmetry [70] which defines systems at equilibrium (see further details in Appendix E.5). This symmetry, implying an infinite hierarchy of fluctuation-dissipation relations, does not hold for the out-of-equilibrium system. This explicitly implies a different number of degrees of freedom in and out-of-equilibrium. In equilibrium, a semi-classical (high temperature) non-relativistic bosonic theory has a term in the action of the form $\varphi_q^* i(\partial_t - \kappa)\varphi_c + c.c. + 4i\kappa T|\varphi_q|^2$, which at a given temperature $T$ imposes a fixed relation between the noise and the loss term (sharing the same $\kappa$). In the non-equilibrium action (15), these two terms are no longer related by a temperature, but they can vary independently, and they undergo different renormalisations.

### 6.5 Comparison of unitary evolution and Lindblad evolution of the NESS

Finally, to get further insights on the spectral function of the driven-dissipative system, we compare the results obtained with the Lindblad evolution (4) to the ones obtained starting from the same $\rho_{\text{NESS}}$, but with unitary evolution (5), in the regime where the coherent parameters are much larger then the incoherent drive and dissipation $J, U \gg 1, z$. The latter can be interpreted as a quenched system: The system is initially open and in the steady state and then, drive and dissipation are turned off and the temporal evolution considered is the unitary one. As shown in Fig. 9 for strong interactions, the spectral functions in both cases closely resemble each other, as one would expect in the limit of vanishing incoherent parameters. The broadening of the lines is very small comparatively to the typical features (depth, width) of the spectrum which are of the order of the coherent parameters.

## 7 Conclusions and perspectives

In this article we have studied the space-time behaviour of two-point correlation functions and the spectral function for a driven-dissipative system of interacting bosons on a lattice,

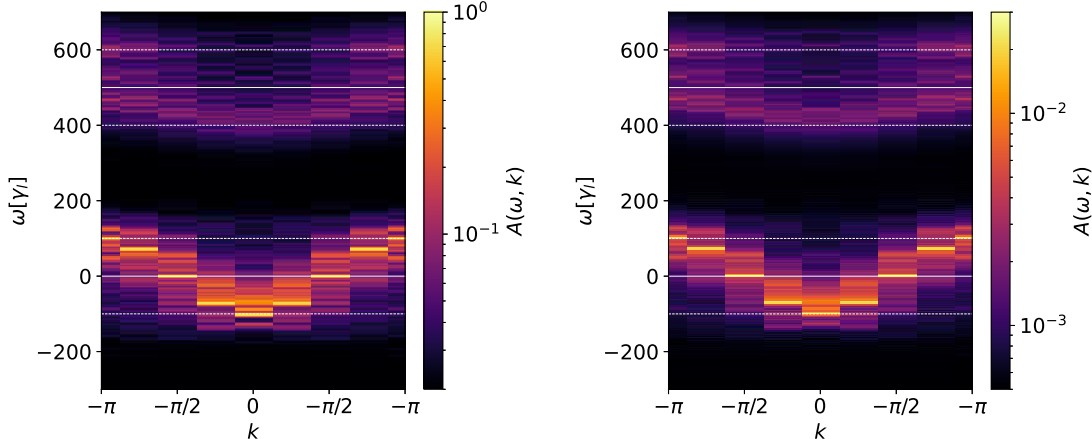

Figure 9: Spectral function $A(\omega, k)$ of the temporal evolution in the NESS. Left: Under unitary dynamics (see Appendix G.5); Right: Under Lindbladian dynamics. The parameters are: $U = 500\gamma_l$, $J = 50\gamma_l$, $z = 0.2$, $L = 8$, $N_s = 3$. The horizontal white lines are as in Fig. 5.

subjected to uniform incoherent pump and one-body losses. Our starting point has been the analytical expression for the NESS density matrix of the system derived in Ref. [68]. We have used several techniques, namely numerical exact diagonalisation, Keldysh formalism and strong coupling approaches, to provide a complete understanding of the excitation spectrum on top of the non-equilibrium steady state. The numerical results that we have obtained cover the fully quantum evolution, allowing us to access in particular the strongly interacting, short distance regime that cannot be reached by field theoretical methods relying on mean-field or perturbative schemes. Our numerical results are confirmed by various exact results in the closed, as well as zero-dimensional system.

We have found that the decay of the retarded Green's function is exponential in time, and the decay rate is renormalised by interactions and increases at both increasing interactions and tunnel amplitude. The spectral function displays at low frequency a quadratic-like branch, very different from the linear branch found for the excitations on top of the ground state of the closed system and for excitations on top of the equilibrium system at low temperature for which we have calculated the spectral function at finite interactions. At large interactions, we have identified, both in the open and in the closed system, a dispersive doublon branch at energy of order $U$. By performing calculations for various system sizes in the closed system case, we have shown that specific features of the spectrum clearly emerging at large system sizes are also recognisable in smaller systems, of typical size accessible via numerical diagonalisation in the open case. By calculating the spectral function for an equilibrium state at finite temperature, we have also highlighted the difference with respect to the out-of-equilibrium one, where, in particular, the linewidth can be tuned by changing the ratio of drive to dissipation rate.

In outlook, on a technical level, it would be interesting to develop new methods to push the calculations for the spectral function of the open system to larger system sizes. This would provide conclusive evidence about the presence of excitations at negative energies (the 'ghost' branches) which, according to Bogoliubov theory, are predicted to be more elusive in the driven-dissipative case than in the closed one [75–77] and could not be resolved in our current numerical calculations. On a fundamental level, our work opens very interesting perspectives for the possibility of observing the effects of interactions in out-of-equilibrium systems within actual experiments both in condensed-matter and

photonic platforms, where short one-dimensional lattices of small photonic resonators can be engineered, and in ultracold atoms in optical lattices, where drive and dissipation are adjustable.

# Acknowledgements

We warmly thank I. Carusotto, L. Garbe and S. Diehl for fruitful discussions, and L. Rosso for significant help in the early stages of the project. MZ thanks for visiting the LPTMS and using their computational infrastructure.

**Funding information**   AM acknowledges funding from the Quantum-SOPHA project ANR-21-CE47-0009; This work is part of HQI (www.hqi.fr) initiative and is supported by France 2030 under the French National Research Agency grant number ANR-22-PNCQ-0002; LM acknowledges funding from the ANR project LOQUST ANR-23-CE47-0006-02; LC and LM acknowledge support from Institut Universitaire de France (IUF).

# A   Detailed characterisation of the non-equilibrium steady state

## A.1   Bosonic NESS density matrix

We show in this section that the Ansatz (9) is a NESS of the Lindblad evolution (2). We abbreviate the weight in each particle sector of the density matrix as $\omega_N = \frac{z^N}{\mathcal{N}}$. The Hamiltonian conserves the particle number and hence commutes with each identity operator in each particle number subspace. Furthermore, the non-hermitian Hamiltonian terms in (2) can be written as

$$\rho_{\text{NESS}} \sum_i b_i^\dagger b_i = \rho_{\text{NESS}} N = N \rho_{\text{NESS}} = \sum_i b_i^\dagger b_i \rho_{\text{NESS}} \, , \tag{31}$$

$$\rho_{\text{NESS}} \sum_i b_i b_i^\dagger = \rho_{\text{NESS}} (N + L) = \sum_i b_i b_i^\dagger \rho_{\text{NESS}} \, , \tag{32}$$

such that we only have to compute the remaining terms

$$\sum_i b_i^\dagger \rho_{\text{NESS}} b_i = \sum_i \sum_N \omega_{N-1} \sum_{\substack{\{l_r\}_{r=1,\dots,L;N} \\ \{n_r\}_{r=1,\dots,L;N}}} |\{l_p\}\rangle\langle\{n_s\}| \sum_{\{m_r\}_{r=1,\dots,L;N-1}} \langle\{l_p\}|b_i^\dagger|\{m_r\}\rangle\langle\{m_r\}|b_i|\{n_s\}\rangle$$

$$= \sum_N \omega_{N-1} \sum_{\substack{\{l_r\}_{r=1,\dots,L;N} \\ \{n_r\}_{r=1,\dots,L;N}}} |\{l_p\}\rangle\langle\{n_s\}|\langle\{l_p\}| \sum_i b_i^\dagger b_i |\{n_s\}\rangle$$

$$= \sum_N N \omega_{N-1} \mathbb{1}_N \, , \tag{33}$$

and similarly,

$$\sum_i b_i \rho_{\text{NESS}} b_i^\dagger = \sum_N (N + L) \omega_{N+1} \mathbb{1}_N \, . \tag{34}$$

The sum of all these contributions vanishes

$$
\begin{aligned}
\mathcal{L}_{\text{loss}}[\rho_{\text{NESS}}] + \mathcal{L}_{\text{gain}}[\rho_{\text{NESS}}] &= \sum_N \mathbb{1}_N \Big[ \gamma_l \Big\{ (N+L)\omega_{N+1} - N\omega_N \Big\} \\
&\qquad + \gamma_p \Big\{ N\omega_{N-1} - (N+L)\omega_N \Big\} \Big] \\
&= \sum_N \mathbb{1}_N \omega_{N-1} \Big[ \gamma_l \Big\{ z^2(N+L) - zN \Big\} + \gamma_p \Big\{ N - z(N+L) \Big\} \Big] \\
&= 0 \;,
\end{aligned}
\tag{35}
$$

which shows that the Ansatz (9) is indeed a steady state.

## A.2 Approach to the NESS

The time evolution of the total particle number $N = \sum_{j=0}^{L} b_j^\dagger b_j$ is given by the Lindblad master equation (2). Since the total particle number commutes with the Hamiltonian $[H, N] = 0$, its evolution is purely generated by the dissipative and pump parts, which yield

$$
\langle \dot{N} \rangle = (-\gamma_l + \gamma_p)\langle N \rangle + \gamma_p L \;.
\tag{36}
$$

Starting with a total occupation of $N_0 := N(t=0)$ and depending on the value of $\gamma_l$ w.r.t $\gamma_p$, we find the following solutions:

- $\gamma_p = \gamma_l$: The particle number grows linearly, there is no steady state. It can be interpreted as a bosonic enhancement, following

$$
\langle N \rangle(t) = \gamma_p L t + N_0 \;.
\tag{37}
$$

- $\gamma_p > \gamma_l$: The particle number grows exponentially, there is no steady state.

- $\gamma_p < \gamma_l$: The particle number changes according to

$$
\langle N \rangle(t) = (N_0 - N_{\text{NESS}})e^{(-\gamma_l + \gamma_p)t} + N_{\text{NESS}} \;,
\tag{38}
$$

where the steady state is given by

$$
N_{\text{NESS}} = \frac{z}{1-z} L \;.
\tag{39}
$$

## A.3 Equal-time correlation functions

We start from equation (11). If the number of particles in position $j$ is $k$, then all other $N - k$ particles are distributed over the remaining $L - 1$ sites. Hence, we have

$$
\begin{aligned}
\sum_{\{m_r\}_{r=1,\ldots,N}} \langle \{m_r\}| b_i^\dagger b_j |\{m_r\}\rangle &= \delta_{ij} \sum_{k=1}^{N} k \left( \binom{L-1}{N-k} \right) \\
&= \delta_{ij} \sum_{k=0}^{N-1} (k+1) \left( \binom{L-1}{N-1-k} \right) \;.
\end{aligned}
\tag{40}
$$

Summing over $N$, we get

$$\sum_{N=1}^{\infty} z^N \sum_{k=0}^{N-1} (k+1) \left( \left( \begin{matrix} L-1 \\ N-1-k \end{matrix} \right) \right) = z \sum_{N=0}^{\infty} z^N \sum_{k=0}^{N} (k+1) \left( \left( \begin{matrix} L-1 \\ N-k \end{matrix} \right) \right)$$

$$= z \sum_{l=0}^{\infty} (l+1) z^l \sum_{M=0}^{\infty} z^M \left( \left( \begin{matrix} L-1 \\ M \end{matrix} \right) \right)$$

$$= z \frac{1}{(1-z)^2} (1-z)^{-L+1}$$

$$= \frac{z}{(1-z)^{L+1}} \, , \tag{41}$$

where from the first to the second line, we used

$$\sum_{p=0}^{\infty} \sum_{q=0}^{p} f(q, p-q) = \sum_{m=0}^{\infty} \sum_{n=0}^{\infty} f(m, n) \, . \tag{42}$$

The geometrical series in the second line only converges if $|z| < 1$. All together, we find the result (12) as expected from the calculation of the total particle number in the NESS.

For the variance, we need to calculate the average over quadratic terms in the local particle number density. Performing similar steps, we obtain

$$\langle n_i^2 \rangle_{\text{NESS}} = \sum_N \frac{z^N}{(1-z)^{-L}} \sum_{\{m_r\}_{r=1,\ldots,N}} \langle \{m_r\} | b_i^\dagger b_i b_i^\dagger b_i | \{m_r\} \rangle$$

$$= \frac{z}{(1-z)^{-L}} \sum_{l=0}^{\infty} (l+1)^2 z^l \sum_{M=0}^{\infty} z^M \left( \left( \begin{matrix} L-1 \\ M \end{matrix} \right) \right)$$

$$= \frac{z(1+z)}{(1-z)^2} \, . \tag{43}$$

### A.4 Oscillation frequency

The dependence of the oscillation frequency on the interaction $U$ can be best understood for the single-particle Green's function under the unitary dynamics in the limit of strong interactions $U \gg J$. We hence consider the evolution under

$$H = \frac{U}{2} \sum_j n_j (n_j - 1) \, . \tag{44}$$

We evaluate the two-point correlation function in the NESS as

$$\left\langle b_j^\dagger(t) b_0 \right\rangle_{\text{NESS}} = (1-z)^L \sum_N z^N \sum_{\{m\}} \langle \{m\} | e^{iHt} b_j^\dagger e^{-iHt} b_0 | \{m\} \rangle \tag{45}$$

$$= (1-z)^L \sum_N z^N \sum_{\{m\}} \langle \{m\} | \left( \prod_k e^{i\frac{U}{2} n_k(n_k-1)t} \right) b_j^\dagger \left( \prod_l e^{-i\frac{U}{2} n_l(n_l-1)t} \right) b_0 | \{m\} \rangle \, .$$

The only non-vanishing contribution is the $j = 0$ one, which is just a consequence of the density matrix being diagonal in the Fock basis. (We can show the same oscillating behaviour for $j \neq 0$ for a more generic density matrix). The matrix element factorises and we find

$$\langle m_0 | e^{i\frac{U}{2} n_0(n_0-1)t} b_0^\dagger e^{i\frac{U}{2} n_0(n_0-1)t} b_0 | m_0 \rangle = m_0 e^{i\frac{U}{2} m_0(m_0-1)t} e^{-i\frac{U}{2}(m_0-1)(m_0-2)t}$$

$$= m_0 e^{iU(m_0-1)t} \, , \tag{46}$$

which indicates an overall oscillation with a period depending on the interaction strength $U$. We finally obtain

$$
\begin{aligned}
\left\langle b_0^\dagger(t) b_0 \right\rangle_{\text{NESS}} &= (1-z)^L \sum_{N=0}^{\infty} z^N \sum_{m_0=0}^{N} m_0 e^{iU(m_0-1)t} \left( \binom{L-1}{N-m_0} \right) \\
&= z(1-z)^L \sum_{N=0}^{\infty} z^N \sum_{m_0=0}^{N} (m_0+1) e^{iUm_0 t} \left( \binom{L-1}{N-m_0} \right) \\
&= z(1-z)^L \sum_{k=0}^{\infty} \sum_{l=0}^{\infty} z^l z^k (k+1) e^{iUkt} \left( \binom{L-1}{l} \right) \\
&= z(1-z) \sum_{k=0}^{\infty} z^k (k+1) e^{iUkt} \;,
\end{aligned}
\tag{47}
$$

with $k$ being the occupation minus one of site zero (relabeled), where we used the infinite double sum rule (42).

## B    Normalisation with truncation of the local Hilbert space

This section gives the normalisation of the NESS density matrix in each $N$-particle sector depending on the system size $L$ and on the local Hilbert space dimension $N_s$. For this, one has to solve a restricted star (balls/particles) and bar (box separations) problem which is well known in combinatorics: How many possible sets $\{x_i\}_{i \in \{1,...,L\}}$ do exist, such that

$$
\sum_{i=1}^{L} x_i = N \,,
\tag{48}
$$

with $x_i \in \{0, ..., N\}$?

The local Hilbert space dimension $N_s$ is equivalent to the existence of an upper bound for the integers $x_i < N_s, \forall i$. Fermions have an upper bound of $N_s = 2$, bosons have no upper bound, but in order to get the correct numerical weight of the state, we need to calculate this problem with an arbitrary upper bound $N_s$. This problem can be solved by using the inclusion-exclusion principle on the lower-bound integer sum. The latter is given by the previous sum with the constraint $x_i \geq a_i$. We substitute $x_i' := x_i - a_i$, such that the modified problem is the unbounded one with $x_i' \geq 0$:

$$
\sum_i x_i' = N - \sum_i a_i \,.
\tag{49}
$$

The number of all possible solutions (as for the bosons) is given by

$$
\binom{L+N-1}{L-1} \,,
\tag{50}
$$

then we subtract

$$
\binom{L}{1} \binom{L+N-1-N_s}{L-1} \,,
\tag{51}
$$

which corresponds to the number of cases where there is at least one box with $x_i \geq N_s$. In the same manner, we construct all cases of at least $k$ boxes with $x_i \geq N_s$ and by adding the

two set intersections, we obtain the required result. The number of possible sets to have in total $N$ identical particles on $L$ distinguishable sites with maximally $N_s - 1$ particles per site is

$$D_{L,N}^{N_s} := \#\text{states}(N_s, L, N) = \sum_{k=0}^{k(N_s-1) \leq N} (-1)^k \binom{L}{k} \binom{L + N - 1 - kN_s}{N - 1} \, . \tag{52}$$

For the special cases of fermions and spin-$\frac{1}{2}$ with $N_s = 2$ and (not truncated) bosons $N_s = \infty$, this simplifies (compare App. C and D to (50)).

## C    Steady-state solution for fermions

We consider the Lindblad master equation (2) where now the annihilation and creation operators are fermionic ones $b_i := c_i$, $b_i^\dagger := c_i^\dagger$, obeying the anti-commutation relations $\{c_i, c_j^\dagger\} = \delta_{ij}$ and $\{c_i, c_j\} = 0 = \{c_i^\dagger, c_j^\dagger\}$. The unitary evolution is given by the Fermi-Hubbard model.

### C.1    NESS density matrix

In finite dimensional Hilbert spaces, there always exists a steady-state solution of the Lindbladian superoperator [52]. Starting from the same Ansatz (9) as in the main text, we obtain from the normalisation of the state that

$$\mathcal{N}_L^F = \frac{1}{(1+z)^L} \, , \qquad z = \frac{\gamma_p}{\gamma_l} \in [0, \infty) \, , \tag{53}$$

where we used that the dimension of the Hilbert space for $N$ fermions in a system of length $L$ corresponds to $D_{L,N}^F = \binom{L}{N}$. This is exactly the size as the unit matrix $\text{Tr}\, \mathbb{1}_N = D_{L,N}^F$. This provides the NESS density matrix for fermions.

### C.2    Equal-time two-point correlation functions

We assume that the sums for fermions run only over non-equal elements. The correlation function is given by

$$
\begin{aligned}
\langle c_i^\dagger c_j \rangle_{\text{NESS}} &= \sum_{N=0}^{L} \frac{z^N}{(1+z)^L} \sum_{\{m_r\}_{r=1,\dots,N}} \langle \{m_r\} | c_i^\dagger c_j | \{m_r\} \rangle \\
&= \delta_{i,j} \sum_{N=0}^{L} \frac{z^N}{(1+z)^L} \binom{L-1}{N-1} \\
&= \delta_{i,j} \frac{z}{1+z} \, ,
\end{aligned}
\tag{54}
$$

where we introduced a complete basis in position space $\{m_r\}$ for $N$ particles and evaluated the sum as

$$
\begin{aligned}
\sum_{\{m_r\}_{r=1,\dots,N}} \langle \{m_r\} | c_i^\dagger c_j | \{m_r\} \rangle &= \sum_{i, \{m_r\}_{r=1,\dots,N-1} \in \{1,\dots,L\} \, \{i\}} \langle \{m_r\} | c_i^\dagger c_j | \{m_r\} \rangle \delta_{i,j} \\
&= \binom{L-1}{N-1} \delta_{i,j} \, .
\end{aligned}
\tag{55}
$$

The variance is given by

$$\langle n_i^2 \rangle_{\text{NESS}} = \sum_N \frac{z^N}{(1+z)^L} \sum_{\{m_r\}_{r=1,\dots,N}} \langle \{m_r\}| c_i^\dagger c_i c_i^\dagger c_i |\{m_r\}\rangle$$
$$= \frac{z}{1+z} \; . \tag{56}$$

Since the number of fermions in state $i$ is the same as its squared value, the result of this operator is the same as the first moment. This leads to

$$\text{VAR}(n_i) = \langle n_i^2 \rangle - \langle n_i \rangle^2 = \frac{z}{(1+z)^2} \; . \tag{57}$$

This function has a maximum variance of $1/4$ for $\gamma_p = \gamma_l$. For either limit of pump much bigger/smaller than the loss, the variance tends to zero.

## D   Steady-state solution for hard-core bosons

We consider a periodic chain of hard-core bosons, which can be mapped by the transformation due to Holstein and Primakoff [78] to the XX spin-$\frac{1}{2}$-chain given by the Hamiltonian

$$H = -2J \sum_{j=1}^{L} (S_j^x S_{j+1}^x + S_j^y S_{j+1}^y) \; , \tag{58}$$

with $S^\mu = \frac{1}{2}\sigma^\mu$ where $\{\sigma^\mu\}_{\mu=x,y,z}$ are the Pauli matrices. We define the first state as the spin-up state $|\uparrow\rangle = |+\rangle = |0\rangle$ and note that the operator $S_j^+ S_j^-$ acts as a local density operator for the spin-up state at site $j$, where $S^+ = S^x + \mathrm{i}S^y$ and $S^- = S^x - \mathrm{i}S^y$, which act as spin-flip operators.

The pump and loss of hard-core bosons hence correspond to spin flips in the XX-model, implying that the resulting Lindbladian is of the form

$$\mathcal{L}[\rho] = -\mathrm{i}[H,\rho] + \gamma_l \sum_{i=1}^{L} \left( S^-{}_i \rho S^-{}_i^\dagger - \frac{1}{2}\{S^-{}_i^\dagger S^-{}_i, \rho\} \right) + \gamma_p \sum_{i=1}^{L} \left( S^+{}_i \rho S^+{}_i^\dagger - \frac{1}{2}\{S^+{}_i^\dagger S^+{}_i, \rho\} \right) . \tag{59}$$

We make the following Ansatz for the non-equilibrium steady state

$$\rho_{\text{NESS}} = \frac{1}{\mathcal{N}_F} \sum_{N=0}^{L} \left( \frac{\gamma_p}{\gamma_l} \right)^N \mathbb{1}_N \; , \tag{60}$$

where the identity operator is the sum over the complete Fock basis for the $N$-particle sector. The number of states in the system of hard-core bosons is identical to the one of the fermionic system, which explains the identical normalisation $\mathcal{N}_F$ and structure of the solution. One can show that the Hamiltonian conserves the particle number, or equivalently the number of up-spins, and hence it commutes with the steady-state density matrix. The dissipative part of the Lindbladian evolution vanishes with the same reasoning as for the bosons and fermions.

## E   Detailed calculations in Keldysh formalism

### E.1   Fourier conventions

We first define the Fourier conventions used in this work. For a lattice of size $L = Na$, where $a$ is the lattice spacing and $N \in 2\mathbb{N}$, we define a set of integers $\mathcal{I} = \{-N/2 +$

$1, \ldots, N/2\}$, such that the discrete positions and discrete momenta are given by

$$x_j = ja \qquad j \in \mathcal{I} \,, \qquad\qquad p_k = k\frac{2\pi}{L} \qquad k \in \mathcal{I} \,. \tag{61}$$

We define the Fourier transforms by

$$f(p_k, \omega) = \sum_{j \in \mathcal{I}} \int_{-\infty}^{\infty} \mathrm{d}t \, f(x_j, t) \, \mathrm{e}^{-\mathrm{i}p_k x_j} \, \mathrm{e}^{\mathrm{i}\omega t} \,, \tag{62}$$

$$f(x_j, t) = \sum_{k \in \mathcal{I}} \frac{a}{L} \int_{-\infty}^{\infty} \frac{\mathrm{d}\omega}{2\pi} \, f(p_k, \omega) \, \mathrm{e}^{\mathrm{i}p_k x_j} \, \mathrm{e}^{-\mathrm{i}\omega t} \,. \tag{63}$$

In the thermodynamic limit $L \to \infty$, the momenta become continuous and the discrete sum tends to continuous integrals

$$\sum_{k \in \mathcal{I}} \frac{1}{N} F(p_k) \to a \int_{-\pi/a}^{\pi/a} \frac{\mathrm{d}p}{2\pi} F(p) \,, \tag{64}$$

for arbitrary functions $F$. Note that, in the main text, as well as in the numerics, we set $a = 1$.

### E.2 Dispersion line and spectral function

We calculate the inverse bare propagator as the second functional derivative of the action evaluated at vanishing field expectation values. Gathering the fields in a vector $\varphi = (\varphi_c, \varphi_c^*, \varphi_q, \varphi_q^*)^T$, this yields

$$\left[G_{0,m\ell}^{-1}\right]_{\alpha\beta}(t, t') \equiv \frac{\delta^2 S[\varphi]}{\delta\varphi_{\alpha,m}(t)\delta\varphi_{\beta,\ell}(t')}\Big|_{\varphi_{c,q}=0} \equiv \begin{pmatrix} \mathbf{0}_{2\times2} & \mathbf{P}_{m\ell}^A(t,t') \\ \mathbf{P}_{m\ell}^R(t,t') & \mathbf{P}_{m\ell}^K(t,t') \end{pmatrix} \,, \tag{65}$$

where the subscripts $m, \ell$ label the discrete position indices and $\alpha, \beta$ the field components. The 0 index in $G_0$ indicates that this is the 'bare' propagator. Since it is evaluated at zero fields, it corresponds to the quadratic (non-interacting) theory. We obtain the inverse Keldysh component as $\mathbf{P}_{m\ell}^K(t, t') = \mathrm{i}\gamma\sigma_x\delta_{m\ell}\delta(t - t')$, and the inverse of the retarded sector of the propagator as

$$\mathbf{P}_{m\ell}^R(t, t') = \begin{pmatrix} 0 & -(\mathrm{i}\partial_t + \mathrm{i}\kappa_0)\delta_{m,\ell} + J_{m,\ell} \\ (\mathrm{i}\partial_t + \mathrm{i}\kappa_0)\delta_{m,\ell} + J_{m,\ell} & 0 \end{pmatrix} \delta(t - t') \,, \tag{66}$$

where $J_{m\ell}$ is a short-hand notation for $J_{m\ell} \equiv (\delta_{m,\ell+1} + \delta_{m,\ell-1})$, and we defined the bare decay rate as $\kappa_0 = (\gamma_l - \gamma_p)/2$ and $\gamma = \gamma_l + \gamma_p$. In order to calculate the excitation branches $\Omega(k)$ in this approximation, we search for the solutions implicitly given by [54, 79]

$$\det \mathbf{P}^R(k, \Omega(k)) = 0 \,, \tag{67}$$

where $\mathbf{P}^R(k, \omega)$ is the Fourier transform in space and time of $\mathbf{P}_{m\ell}^R(t, t')$ defined according to (62). Note that it only depends on one momentum and one frequency because of translational invariance in space and time. From (66) we find

$$\Omega_\pm(k) = -\mathrm{i}\kappa_0 \mp 2J\cos(ka) \,, \tag{68}$$

which is the solution given in the main text with a lattice spacing $a = 1$.

### E.3   Occupation number

The average occupation number $\langle n_j \rangle$ is encoded, within the Keldysh formalism, in the Keldysh Green's function $G^K$, which is related to the second quantization operators by

$$G_{jj}^K(t,t) = -\mathrm{i}\Big\langle \{b_j(t), b_j^\dagger(t)\} \Big\rangle = -\mathrm{i}(2\langle n_j \rangle + 1)\,. \tag{69}$$

By inverting the inverse bare propagator $G_0^{-1}$, one obtains the bare Keldysh component $G_0^K$ defined in (75) as

$$G_{jj,0}^K(t,t) = a \int_{-\pi/a}^{\pi/a} \frac{\mathrm{d}p}{2\pi} \int_{-\infty}^{\infty} \frac{\mathrm{d}\omega}{2\pi} \frac{-\mathrm{i}\gamma}{(\omega + 2J\cos(pa))^2 + \kappa_0^2} = -\mathrm{i}\frac{\gamma}{2\kappa_0} = -\mathrm{i}\frac{\gamma_l + \gamma_p}{\gamma_l - \gamma_p}\,, \tag{70}$$

from which one deduces

$$\langle n_j \rangle \equiv \langle n \rangle = \frac{1}{2}\left(\frac{\gamma_l + \gamma_p}{\gamma_l - \gamma_p} - 1\right) = \frac{z}{1-z}\,, \tag{71}$$

which does not depend on space as expected for a homogeneous system. This coincides with the result (12).

### E.4   Perturbative corrections

We now turn to the calculation of the one-loop correction to the self-energy defined in (20). First, we introduce into the partition function $Z$ a source term $J = (J_q^*, J_q, J_c^*, J_c)^T$ linearly coupled to the field $\varphi$ as

$$Z[J] = \mathrm{e}^{\mathrm{i}W[J]} = \int \mathcal{D}\varphi \, \exp\left\{ \mathrm{i}S[\varphi] + \mathrm{i}\int \mathrm{d}t \sum_j \varphi_j^T(t)J_j(t) \right\}\,. \tag{72}$$

The Schwinger functional $W[J]$ is the generating functional of $n$-point connected correlation functions $G^{(n)}$, which are defined as

$$G_{a_1\ldots a_n}^{(n)} \equiv \prod_{k=1}^{n} \left(\frac{\delta}{\mathrm{i}\delta J_{a_k}}\right) W[J]\Big|_{J=0}\,, \tag{73}$$

where, to alleviate notations, we employ from now on de Witt convention, whereby the roman letter indices stand for the field components and their respective spacetime arguments. In particular, the two-point connected Green's functions $G_{ab}^{(2)}$ are given by

$$G_{ab}^{(2)} \equiv -W_{ab}^{(2)}[J=0] = \left(\frac{\delta^2}{\mathrm{i}\delta J_a \mathrm{i}\delta J_b}\right) W[J]\Big|_{J=0}\,. \tag{74}$$

They can be gathered in a $4 \times 4$ matrix as

$$G^{(2)} = \begin{pmatrix} \mathbf{G}^K & \mathbf{G}^R \\ \mathbf{G}^A & \mathbf{0}_{2\times 2} \end{pmatrix}\,, \tag{75}$$

where the $2 \times 2$ matrices $\mathbf{G}$ are named Keldysh $\mathbf{G}^K$, advanced $\mathbf{G}^A$ and retarded $\mathbf{G}^R$ propagators, and are constrained by causality as $\mathbf{G}^A(\omega) = (\mathbf{G}^R(\omega))^*$.

Finally, we define the effective action $\Gamma$ as the Legendre-Fenchel transform of $W$ w.r.t the one-point expectation value of the fields $\phi \equiv \langle \varphi \rangle_J$

$$\Gamma[\phi] \equiv \inf_{J[\phi]} \left( W[J] - \sum_a J_a \phi_a \right)\,, \tag{76}$$

where the sum stands for summation over field and source components, as well as summation/integration over their spacetime arguments. For a system at equilibrium, the effective action corresponds to the Gibbs free energy and the correlation functions obtained by taking functional derivatives of $\Gamma$ with respect to $\phi$

$$\Gamma^{(n)}_{a_1 \ldots a_n}[\phi] \equiv \frac{\delta^n}{\delta\phi_{a_1} \ldots \delta\phi_{a_n}} \Gamma[\phi] \,, \tag{77}$$

correspond to the renormalised vertex functions of the theory. From the Legendre relation, one deduces that the matrix of second derivatives of the effective action is the inverse of the propagator matrix $G^{(2)}$

$$G^{(2)} = \left[\Gamma^{(2)}\right]^{-1} \,. \tag{78}$$

In order to obtain the self-energy, following usual field-theoretical methods, we compute $\Gamma^{(2)}$ within a one-loop expansion at small interaction. At one-loop order, it is given by

$$\Gamma_{(1\text{-loop})} = S - \frac{\mathrm{i}}{2} \operatorname{Tr} \ln S^{(2)} \,, \tag{79}$$

where the trace means again summation over the fields and summation/integration over their spacetime arguments. Taking two functional derivatives of this expression with respect to the fields, one obtains

$$\left[\Gamma^{(2)}_{ab}\right]_{(1\text{-loop})} = G^{-1}_{0,ab} - \frac{\mathrm{i}}{2} \frac{\delta^2}{\delta\phi_a \delta\phi_b} \operatorname{Tr} \ln S^{(2)} \,. \tag{80}$$

Identifying with the definition (20) for the self-energy yields

$$\left[\Sigma_{ab}\right]_{(1\text{-loop})} = \frac{\mathrm{i}}{2} \frac{\delta^2}{\delta\phi_a \delta\phi_b} \operatorname{Tr} \ln S^{(2)} \,. \tag{81}$$

This can be represented in terms of Feynman diagrams. With a four-point vertex $(-U)$ and propagator lines corresponding to the Green's function components, the only diagrams one can construct are tadpole diagrams. Because of the integral over the frequency in the loop, the only non-vanishing contributions stem from diagrams constructed with the Keldysh propagator. This is a consequence of causality: The advanced and retarded propagators only have poles in a half complex plane and vanish upon integration on frequency. One finds

$$\Sigma_{(1\text{-loop})} = \begin{pmatrix} 0 & 0 & 0 & \Sigma_1 \\ 0 & 0 & \Sigma_1 & 0 \\ 0 & \Sigma_1 & 0 & 0 \\ \Sigma_1 & 0 & 0 & 0 \end{pmatrix} \,, \tag{82}$$

where $\Sigma_1$ is given by

$$\Sigma_1 = \frac{\mathrm{i}}{2} \times 2 \times (-U) \int_{-\pi/a}^{\pi/a} a\frac{\mathrm{d}p}{2\pi} \int \frac{\mathrm{d}\omega}{2\pi} \frac{-\mathrm{i}\gamma}{(\omega + 2J\cos(pa))^2 + \kappa_0^2} = \frac{\gamma U}{2\kappa_0} \,, \tag{83}$$

where the factor 2 takes the multiplicity of the diagram into account, the factor $-U$ arises from the 4-point vertex, and the Keldysh propagator forming the loop is integrated over.

## E.5 Time-reversal symmetry

A (quantum) system at equilibrium is invariant under time reversal. This definition of equilibrium manifests in the Keldysh action as an invariance under a discrete symmetry transformation [70]. Considering the entire quantum evolution, it is easy to verify that the action (15) does not fulfill the time reversal symmetry transformation, as the quadratic part in the quantum fields does not contain the term $\coth \beta\omega$, relating it to the Bose-Einstein distribution $n(\omega) = 1/(\mathrm{e}^{\beta\omega} - 1)$.

Also, in the semi-classical limit, *i.e.* in the limit of large temperature $T$, this discrete transformation simplifies to

$$\begin{cases} \mathcal{T}\Phi_{c,j}(t) &= \sigma_x \Phi_{c,j}(-t) \\ \mathcal{T}\Phi_{q,j}(t) &= \sigma_x \left( \Phi_{q,j}(-t) + \frac{\mathrm{i}}{2T}\partial_t \Phi_{c,j}(-t) \right), \end{cases} \tag{84}$$

where $\Phi_\nu$ denotes the vectors $\Phi_{\nu,j}(t) \equiv (\varphi_{\nu,j}(t), \varphi^*_{\nu,j}(t))^T$ with $\nu = c, q$. This transformation is equivalent to the well-known relation for classical dynamical equilibrium systems in the Martin-Siggia-Rose formalism [80, 81]. On can check that the action (15) is not invariant under this transformation. Indeed, the terms in the first line of (15) can only be time-reversal symmetric in the presence of the additional coherent contribution

$$\int \mathrm{d}t \sum_j \kappa \left( \varphi^*_{q,j} \, \partial_t \varphi_{c,j} - \varphi^*_{c,j} \, \partial_t \varphi_{q,j} \right), \qquad \text{with } \kappa = -\frac{\gamma}{4T}$$

and with $\gamma_l = \gamma_p$, *i.e.* $\kappa_0 = 0$. Hence, for $\kappa = 0$ and $\kappa_0 \neq 0$, the system breaks time-reversal symmetry, it is genuinely out-of-equilibrium.

## F Decay rate at large interactions

For $J = 0$, each site decouples. Hence, in the strong interaction limit $U \to \infty$, one is left with solving the problem of $L$ independent dissipative Kerr resonators. This is a well-known exactly solvable problem [69]. One can exploit a weak symmetry to block-diagonalise the Lindbladian and then introduce an operator which renders the Lindbladian quadratic in the creation and annihilation superoperators, i.e. the operators acting on density matrices. Equivalently, one can introduce a time-dependent gauge transformation to obtain a quadratic action as shown in [59]. The Kerr non-linearity then transforms into a fluctuating frequency depending on the number density which in turn leads to dephasing. The result for the retarded Green's function is given by [69]

$$G^R(t) = -\mathrm{i}\Theta(t) \frac{\mathrm{e}^{\mathrm{i}Ut + \frac{\gamma_l - \gamma_p}{2}t}}{\left( \cosh \frac{\Gamma}{2}t + R_1 \sinh \frac{\Gamma}{2}t \right)^2}, \tag{85}$$

where

$$\Gamma := \sqrt{(\gamma_l - \gamma_p)^2 - U^2 + 2\mathrm{i}U(\gamma_l + \gamma_p)}, \qquad R_1 := \frac{1}{\Gamma}\left[ (\gamma_l - \gamma_p) + \mathrm{i}\frac{U(\gamma_l + \gamma_p)}{\gamma_l - \gamma_p} \right]. \tag{86}$$

We can expand the term (assuming $t > 0$)

$$\begin{aligned} \kappa_\infty t &= -\Re\left[ \ln \mathrm{i}G^R(t) \right] \\ &= -\Re\left[ \mathrm{i}Ut + \frac{\gamma_l - \gamma_p}{2}t \right] + 2\Re \ln \left[ \cosh \frac{\Gamma}{2}t + R_1 \sinh \frac{\Gamma}{2}t \right] \\ &= -\frac{\gamma_l - \gamma_p}{2}t + t\Re\Gamma + 2\Re \ln \left[ 1 + R_1 + \mathrm{e}^{-\Gamma t}(1 - R_1) \right]. \end{aligned} \tag{87}$$

The last term is suppressed in the large $t$ limit. Expanding the second term for $U \gg \gamma_l$, we find that

$$\Re\, \Gamma = \gamma_l + \gamma_p + \mathcal{O}(U^{-2}) \,. \tag{88}$$

such that the renormalised decay for $U \gg \gamma_l$ reads

$$\kappa_\infty = \frac{\gamma_l + 3\gamma_p}{2} = \kappa_0 + 2\gamma_p \,. \tag{89}$$

## G    Details on the numerical calculations

### G.1    Exact diagonalisation

Numerical results were obtained by exact diagonalisation of the Hamiltonian or Lindbladian in the Fock basis of particle occupation numbers per site and the time evolution implemented in the qutip library [82, 83]. In the case of the closed-system evolution, we additionally use the conservation of particle number, as explained in section G.5.

### G.2    Fit of the decay

In order to extract the decay of the retarded Green's function (3) in the NESS, we assume that it is of the form given by the single-pole Ansatz (24) and define

$$\mathcal{G}(t) := -\Re \ln \mathrm{i} G_{00}^R(t) \,. \tag{90}$$

In the non-interacting regime, the single-pole Ansatz is exact and one has $\mathcal{G}(t) = \kappa t$. In the interacting case, the decay is renormalised and we determine it from the linear trend of the oscillating curve $\mathcal{G}(t)$.

Restricting the dimension of the local Hilbert space, we obtain results for all interaction strength by exact diagonalisation. Using this method, we extract the slope of $\mathcal{G}(t)$ by a linear fit to the oscillation curve. An example can be found in Fig. 10. Due to the oscillations on different time scales – note that we vary both $U$ and $J$ over several orders of magnitude –, the result for $\kappa$ can depend on the time intervals chosen. In turn, fitting a family of 20 curves in different time intervals gives an estimate of the variance of the fitted decay, which is the dominant error source of the fit.

### G.3    Effect of the Hilbert space truncation

In the main text we showed that in the case $J = 0$ we recover the analytically predicted renormalisation of the decay $\kappa(U)$. In order to benchmark the numerical truncation over the local Hilbert space, we here investigate the dependence of $\kappa(U)$ for large hopping values, taking $J/\gamma_l = 10$. We show in Fig. 11 the analytical (single-site) prediction, holding in the limit of vanishing $J$, and the numerical results for $N_s \in \{3, 4, 5\}$. For large $U$, we expect a deviation from the single-site result and we observe that the reached plateau is independent of the local Hilbert space truncation. In the weak interaction limit, we interpret the closeness of the results for $N_s = 4$, $N_s = 5$ with the exact prediction as an indication of numerical convergence.

### G.4    System size dependence

We further investigate the dependence of the reached plateau value of the decay for large $U$ and $J$ on the system size $L$. From Fig. 11, we conclude that a truncation of $N_s = 3$ is

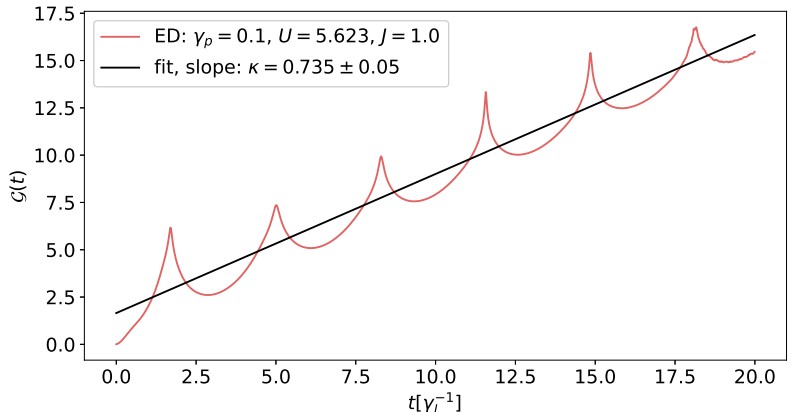

Figure 10: Logarithm of the retarded Green's function $\mathcal{G}(t) = -\Re\left[\ln \mathrm{i} G_{00}^R(t,0)\right]$ (dimensionless) obtained from exact diagonalisation (red solid line), as a function of time (in units of $\tau_l$) for the site $j = 0$. Within the single-pole approximation, the decay rate is estimated by fitting $\mathcal{G}(t)$ by a linear model. The result of the fit is shown as the black solid line. The parameters are $U = 5.623\gamma_l$, $J = \gamma_l$, $z = 0.1$, $L = 4$, $N_s = 4$.

sufficient at large $U$ to converge numerically at the correct plateau value. Hence, we fix $N_s = 3$ and vary the system size $L$. The dependence of the renormalised decay $\kappa$ on $L$ is shown in Fig. 12, in a limit of large hopping and interaction (numerical values: $J/\gamma_l = 10$ and $U/\gamma_l = 100$). We observe that the effective decay increases monotonously with the system size for these small system sizes considered.

## G.5 Exact diagonalisation for the unitary system

We calculate the Green's function as the overlap of two time-evolved states. Since we are interested in the unitary time evolution, the dimensionality of the problem greatly reduces, and is further diminished by the symmetries present in the problem (particle number conservation, momentum conservation and spatial parity conservation). We calculate the Green's function as

$$G_{j0}^R(t,0) = -\mathrm{i}\Theta(t)\mathrm{Tr}\left\{\left[b_j(t), b_0^\dagger(0)\right]\rho_{\mathrm{NESS}}\right\} \tag{91}$$

$$= -\mathrm{i}\Theta(t)\left(G_j^{(1)}(t) - G_j^{(2)}(t)\right) \tag{92}$$

with

$$G_j^{(1)}(t) := \mathrm{Tr}\left\{\mathrm{e}^{\mathrm{i}Ht} b_j \mathrm{e}^{-\mathrm{i}Ht} b_0^\dagger \rho_{\mathrm{NESS}}\right\}$$

$$= \mathcal{N}^{-1} \sum_{N=0} z^N \sum_{\{m_i\}_N} \mathrm{Tr}\left\{b_j \left|\mathrm{e}^{-\mathrm{i}H_{N+1}t} b_0^\dagger \{m_i\}_N\right\rangle \otimes \left|\mathrm{e}^{-\mathrm{i}H_N t} \{m_i\}_N\right\rangle^\dagger\right\}, \tag{93}$$

$$G_j^{(2)}(t) := \mathrm{Tr}\left\{b_0^\dagger \mathrm{e}^{\mathrm{i}Ht} b_j \mathrm{e}^{-\mathrm{i}Ht} \rho_{\mathrm{NESS}}\right\}$$

$$= \mathcal{N}^{-1} \sum_{N=0} z^N \sum_{\{m_i\}_N} \mathrm{Tr}\left\{b_j \left|\mathrm{e}^{-\mathrm{i}H_N t} \{m_i\}_N\right\rangle \otimes \left|\mathrm{e}^{-\mathrm{i}H_{N-1}t} b_0 \{m_i\}_N\right\rangle^\dagger\right\}. \tag{94}$$

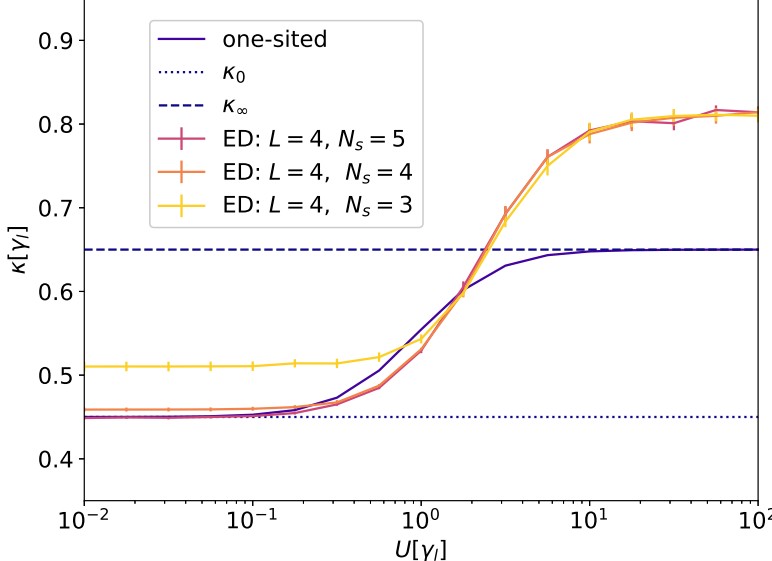

Figure 11: Decay $\kappa(U)$ for $J = 10\gamma_l$, $z = 0.1$, and $L = 4$, and for different local truncation of the Hilbert space $N_s$ in the numerics. We expect to recover the non-interacting solution for small $U$ and a deviation from the single-site solution in the limit of large interaction.

In order to reduce the Hilbert space dimension further, we construct the Hamiltonian in each particle number sector, since $[N, H] = 0$. Then, we draw states from the distribution $\rho_{\text{NESS}}$, which constitute a sequence of numbers allowed by the local Hilbert space cutoff, and evolve them with the block Hamiltonian constructed in the reduced basis. This is performed using qutip [82, 83], which optimises the temporal evolution. We add and subtract particles (before/ after the evolution) by action on the sequence directly.

### G.6    Simulations with matrix product states

In order to produce the plots in Fig. 6 we employ a representation of the many-body state as a matrix-product state [84] and code our algorithm using the Itensors library [85, 86]. We prepare the ground state considering a maximum number of bosons per site equal to 2 on a lattice with periodic boundary condition and at density $N/L = 1/4$. We subsequently compute the lesser and greater Green's functions $G_{0j}^<(t) = -i\langle b_j^\dagger(t)b_0\rangle$ and $G_{0j}^>(t) = -i\langle b_0 b_j^\dagger(t)\rangle$ and using these data we reconstruct the spectral functions plotted in the figure. For the time evolution, we fix $J = 1$ and use a time step of $\delta t = 0.01$. The chemical potential is obtained from the ground state energy $E_N$ of a $N$-particle system by taking the discrete derivative $\mu = E_{N+1} - E_N$, where $E_N$ is obtained from the DMRG calculations.

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
