# Peer review of "Space-time first-order correlations of an open Bose Hubbard model with incoherent pump and loss"

_SciPost Physics_

## Round 1 · Referee Report · Anonymous (Referee 1) · 2024-7-9

Strengths

1- The steady state of this model was already know to be the infinite temperature state with a given fugacity. The novelty of this paper is the study of the first order correlations. The system is studied both with exact diagonalization (ED) and with the Keldysh field formalism. It is only within the ED approach that new results on the correlations are obtained.

2- The paper is well written and easy to follow. The introduction gives a good overview of the field of open many body systems and the theoretical techniques to deal with them.

Weaknesses

1- The problem I have with the manuscript is related to the perspective the results are put in. In the abstract and conclusions, the difference with the correlations in the ground state of the Bose-Hubbard model are emphasised. It is a bit a strange choice to put the emphasis on these differences, because the steady state is rather the infinite temperature one. Actually in Sec. 5.4 the comparison with the infinite temperature closed system is made and a good correspondence is observed. At least, this is what I get from Fig. 7 and the describing text. It is said in the abstract that “the dispersion is quadratic instead of linear at small wave vectors”. But in Fig. 7, the dispersion also looks quadratic to me for the closed system.
I would therefore tend to say that the driving and dissipation has a small effect on the first-order correlations. I agree that this conclusion may be a bit disappointing and that the authors therefore put the emphasis on the difference with the zero-temperature state. But that conclusion would also hold for comparing the infinite and zero temperature closed systems and no one would be surprised by that.
Given the close similarities between the left and right hand panels in Fig. 7, I could imagine that it is possible to develop a perturbative treatment of the driving and dissipation, but it is hardly the task of the referee to suggest new research.

Report

The manuscript by Zündel et al. presents a theoretical analysis of the first order correlation function of a driven-dissipative Bose-Hubbard model with incoherent pumping and losses.

The current manuscript leaves me a bit wondering whether there is something interesting to be said about the first order coherence for this system with respect to the closed system. This leaves one with a nice piece of work where the conclusions seem to be unfortunately a bit trivial. But of course the authors are very much welcome to point out my oversights.

Requested changes

1- Unless I have missed it, the manuscript actually does not point out a big impact of the nonequilibrium condition on the first order correlation function. If the authors cannot convince me that I am wrong, I am afraid that the manuscript lacks a clear message that warrants publication in SciPost.

2- As a side remark, I would like to mention that I did not appreciate very much the discussion of the work in perspective of the NISQ era in the conclusions. There does not seem to be any potential of the presented driven-dissipative Bose-Hubbard model in this respect. It is my opinion that these boiler plate texts should be avoided as much as possible in the ChatGPT era.

Recommendation

Ask for major revision

---

## Round 1 · Referee Report · Anonymous (Referee 2) · 2024-7-15

Strengths

- Pedagogical introduction
- High clarity of discussion
- High degree of experimental relevance
- Simple and generic model which is widely studied and of interest to a broad community of researchers
- Rigorous comparison to a number of limiting cases and a direct connection to ground-state results, for which readers may be more familiar.

Weaknesses

- Direct connection to experiment could be better reflected in references.
- There is little discussion of why the Green's function is expected to have no interaction dependence.

Report

The authors consider the Bose-Hubbard model with uniform gain and loss in a Lindblad master equation. Numerical analysis of the full system is complemented by analytical results in limiting cases. After the first two introductory sections, the steady state itself is discussed and its simple nature is emphasized. The retarded Greens function are computed and compared to results for ground-states, as well as to unitary evolution from the steady state. It is demonstrated that dissipation alters the low energy excitations. It is also shown that despite the fact that the steady state has no signature of the interactions, the relaxation towards this state has a clear dependence on the interactions.

The paper is highly pedagogical and easy to follow, with an intuitive and comprehensive introduction. The techniques employed are appropriate and I believe the results are of significant value to the community given current experimental interest in relatively small Bose-Hubbard arrays. Specialists will appreciate the details, while non-specialists should also be able to follow the clear exposition. The comparison between the dissipative spectral function and that of the ground-state is particularly interesting and clear, emphasising how dissipation changes the nature of excitations. This should assist the reader to make a connection to more familiar results for closed quantum systems.

Specific Comments:

⁃ Overall, I think the direct connection to experiment could be more strongly emphasized by providing a few more references. There is a sentence “While all these studies concern closed quantum systems, in several experimental situations quantum systems are subjected to external pump and/or losses, or put in contact with some type of bath.” Following this, there is a list of “examples of bosonic open systems”, however, only a few experiments are listed. Given the preceding sentence, I was expecting a list of experiments. This is particularly true for [31]-[40]. A few more experimental references here would be appreciated. Possible examples that are not currently present are experiments with lossy Bose-Hubbard arrays in which a subsystem experiences an effective (though possible coherent) gain due to transport from other parts of the lattice. An example can be found in PRL 116, 235302 (2016) and there are many examples of non-equilibrium Bose-Hubbard experiments in the review AVS Quantum Sci. 3, 039201 (2021). These could be used to add to the broader context of this work.

⁃ There are other models without any interaction dependance in the steady states, but non-trivial fluctuations and decay that depend on interactions. However, to my knowledge these are critical systems with dark states, specifically, absorbing dark states where the steady state is a vacuum. For example, PRA 98, 062117 (2018) and PRL 132 120401 (2024). The manner in which the Bose-Hubbard example here is ‘featureless’ is of course different, being an infinite temperature steady state with a given fugacity. Presumably there are simple counter-examples where the steady state is trivial but there is no signature of interactions in the spectral function? Generically, dephasing leads to infinite temperature states. In those cases, would there be any signatures of interactions in the approach to the steady state? Since the authors emphasise that this aspect is “remarkable”, it would be nice to have some discussion of the broader expectations here. Tangential to this, why is it surprising that the decay rate and oscillation frequency would be renormalized by the interactions?

⁃ Fig 3a: A lot of negative times are shown in the figure. Is there a reason for this? It is a little bit distracting and some space could be saved by reducing this.

⁃ Fig 3b: Are the oscillatory features best shown in the log-plot? Perhaps this could be clarified with an inset? I appreciate that the decay may make it difficult to visualise with a linear scale and the current figure may be as clear as it can be.

⁃ Fig 4. I think the color bar label is missing.

Pending clarifications to my questions and subsequent minor changes where appropriate, I am happy to recommend this for publication in SciPost Physics.

Requested changes

1. Improved connection to experiment

2. Better discussion around why it is remarkable that the spectral function has interaction dependence, despite the absence of interaction effects in the steady state.

Recommendation

Ask for minor revision

---

## Round 1 · Referee Report · Anonymous (Referee 3) · 2024-7-18

Strengths

1- The paper presents a thorough analysis of correlations in the steady state of a driven-dissipative Bose-Hubbard model, combining insights from different analytical approaches with numerics.

2- The paper shows that even though the steady state is at infinite temperature, temporal correlations reveal nontrivial dynamics

Weaknesses

1- Given that the steady state is at infinite temperature, the ground state of the closed system is not a well-motivated point of reference. In particular, the steady state is not continuously connected to the ground state in the limit $\gamma_p, \gamma_l \to 0$ at fixed ratio $z$.

2- The comparison to the isolated system at infinite temperature in Fig. 7 shows almost identical behavior. Therefore, it is not clear whether the "features of the spectral function are a clear signature of the non-equilibrium nature of the quantum system" as claimed in the introduction.

Report

The paper studies correlations in the steady state of a driven-dissipative Bose-Hubbard model, using a combination of analytical and numerical approaches. The presentation is clear and the results are interesting. In particular, the nontrivial dependence of correlations and spectral features on hopping and interactions, even though the system is essentially at infinite temperature, is perhaps unexpected.

However, I do not think that the claim that the "features of the spectral function are a clear signature of the non-equilibrium nature of the quantum system" is justified. On the one hand, the comparison to the isolated system at zero temperature seems a bit far-fetched; on the other hand, the comparison to the isolated system at infinite temperature shows almost identical features. Therefore, what the consequences of the system being driven and dissipative are remains somewhat unclear.

Moreover, at the moment I do not see how the paper satisfies the criteria of providing "a novel and synergetic link between different research areas," or of opening "a new pathway in an existing or a new research direction." Therefore, I do not think that the paper is suitable for publication in SciPost Physics, at least not in its present form.

Requested changes

1- In Fig. 5, on the left-hand side there seem to be $L = 8$ different values of momenta, but on the right-hand side the values of $k$ look continuous. Why is that the case? Further, for $N_s = 3$, there should be three white lines in the plot on the left-hand side, but I can see only two.

2- What are the blue, red, and green lines in Fig. 6?

3- What is the exact solution shown in the left panel in Fig. 6? How is $\mu$ determined?

4- For small system sizes as shown in Fig. 5 and the left panel of Fig. 6, the difference between a linear and a quadratic dispersion is really hard to see. Is there a better way to show this?

5- In the captions of some figures (e.g., Fig. 7, 8, 9, 10) parameters are not given in units of $\gamma_l$.

6- Do the right panels of Figs. 5 and 7 show the same data? This is somewhat confusing.

7- What is the meaning of the symbol $\omega_{N - 1, L}$ in Eq. (30)?

8- The discussion in Appendix B is unclear. What is the "problem" defined by Eq. (45)? What are solutions to that problem? What determines the value of the upper bound $d_l$? What happens going from Eq. (47) to (48)?

9- It is not quite clear to me why Appendices C and D are included in the manuscript.

10- What is the parameter $\gamma$ in Eq. (50)?

11- I believe in Eq. (52) there is a typo: it should be $\{1, \dotsc, L \} \setminus \{ i \}$.

12- What are the "superoperators" mentioned in Appendix E?

13- Equations (61) and (62) are supposedly expansions in $U \gg J$, but $J$ does not appear in these equations.

14- The discussion of the fitting procedure in the second paragraph of Sec. F.2 is unclear.

15- The caption of Fig. 8 refers twice to the black solid line, which I find somewhat confusing.

16- I do not think that "one-sited" in Sec. F.3 is a commonly used word. What is meant here? Is this the result for a single site? But this would correspond to $J = 0$ and not to small $J$ as stated in the text.

17- What is the parity that is conserved as stated in Sec. F.5?

18- Where is the approach described in Sec. F.5 used? I suppose it is used to obtain the left panel in Fig. 7, but this could also be obtained by using the same approach as for the right panel, simply by setting $\gamma_l = \gamma_p = 0$ in the time evolution.

19- In Fig. 10, it seems surprising that the results for $L = 2, 4$ agree but are way off for $L = 6$. Why is that? In the same figure, what is the meaning of the error bars?

20- The Heaviside function is denoted by both $\theta$ and $\Theta$.

Recommendation

Ask for major revision

---

## Round 2 · Referee Report · Anonymous (Referee 1) · 2025-1-14

Report

The authors have answered in detail and convincingly to all the issues that were raised in my report and in the reports of the other referees. I therefore believe that the manuscript in its current form is suitable for publication in SciPost.

Recommendation

Publish (meets expectations and criteria for this Journal)

---

## Round 2 · Referee Report · Anonymous (Referee 2) · 2025-2-3

Strengths

- Pedagogical introduction
- High clarity of discussion
- High degree of experimental relevance
- Simple and significant observations about a generic model which is of interest to a broad community of researchers
- Rigorous discussion of a number of limiting cases and a direct comparison to ground-state and equilibrium results, for which readers may be more familiar.

Report

The manuscript utilizes a host of techniques to provide a comprehensive analysis of dynamical properties and correlations for the steady-state of the Bose-Hubbard model with uniform gain and loss, a paradigmatic driven-dissipative system with significant experimental relevance. 
 The authors make a number of observations, including demonstrating that the particle lifetime is renormalized with interaction strength and hopping amplitude, that the retarded Greens function oscillates at a frequency set by the interaction strength, and that the spectral function contains signatures of interactions and has a tunable line width. 
The extensive discussion of the corresponding ground state and thermal properties allows the authors to succinctly highlight the nontrivial features of the system. The concluding discussion appropriately identifies future directions and emphasises the current technical barriers, such as the absence of numerical techniques that can be applied to large-scale systems.

The new submission is a significant improvement. In particular, the connection to experiment is better reflected in the references, and the effect of interactions on the retarded Greens function is treated more comprehensively. The discussion of 1-loop corrections due to the interactions is particularly appreciated and allows for a much more direct connection with the Keldysh field theory literature. The numerics also illustrate that the decay rate is renormalized by interactions, which is not captured in the perturbative 1-loop calculation.

Given the above, the manuscript fulfils the journal criteria of "Open a new pathway in an existing or a new research direction, with clear potential for multi-pronged follow-up work". The work may also fulfill a second criteria: "Provide a novel and synergetic link between different research areas" - the connections with equilibrium and ground-state properties combined with the mix of analytical and numerical methods certainly provides synergetic links, and for this system it is also novel to my knowledge. The authors have sufficiently addressed my criticisms from the first review and I believe the paper is appropriate for publication in SciPost Physics as is. Below, I have listed minor comments/suggestions which potentially could improve the manuscript (at the author's discretion).

For the comparison to the equilibrium case: 

1) if the equilibrium spectrum is linear at low energy could this difference also be highlighted in the conclusion? 
For instance, it currently states “The spectral function displays at low frequency a quadratic-like branch, very different from the linear branch found for the excitations on top of the ground state of the closed system, for which we have calculated the spectral function at finite interactions.”

2) Doesn't the equilibrium case also have a doublon branch at high temperatures? If so, "remarkable" in the following statement could perhaps be tempered slightly: “Second, we identify an additional excitation branch centered around energy U , which is the analogue of the doublon branch of the closed-system case. This is remarkable as ρNESS does not contain information on the interaction itself and does not depend explicitly on U.” 

Finally, I am curious, does the one-loop calculation get the oscillation frequency of the retarded Greens function correct, or give a reasonable approximation? There are comments about the decay rate being renormalized but it wasn't clear what happened to the oscillation frequency.

Requested changes

None

Recommendation

Publish (meets expectations and criteria for this Journal)

---

## Round 2 · Referee Report · Anonymous (Referee 3) · 2025-2-7

Report

The authors have addressed my concerns and the points raised by the other referees convincingly. The newly added results sharpen the message of the paper considerably. In particular, the comparison to a system in thermal equilibrium at high temperature illustrates clearly which novel features can be seen out of equilibrium. I also appreciate the clarifications concerning the acceptance criteria of SciPost Physics. I recommend publication of the manuscript in its present form.

Recommendation

Publish (meets expectations and criteria for this Journal)

---

## Round 2 · Author Response

Errors in user-supplied markup (flagged; corrections coming soon)

“Space-time first-order correlations of an open Bose Hubbard model with incoherent pump and loss”
by M. Zündel et al.

Dear Editors,

we would like to resubmit our paper titled “Space-time first-order correlations of an open Bose Hubbard model with incoherent pump and loss” to SciPost Physics. We thank the Referees for the careful reading and the very useful comments on our paper, which have helped us to improve the manuscript. As suggested by the Referees, we did a major revision of the paper, adding calculations, figures and we also shifted the focus of the paper. Please find below the detailed answers to all the points raised. We have taken them into account in the revised version of our manuscript, where major changes are highlighted in red. We hope that the paper is now suitable for publication in SciPost Physics.

With best regards,
Martina Zündel and Anna Minguzzi on behalf of the authors

=====================================================
Answer to the First Referee
=====================================================

Strengths

1- The steady state of this model was already know to be the infinite temperature state with a given fugacity. The novelty of this paper is the study of the first order correlations. The system is studied both with exact diagonalization (ED) and with the Keldysh field formalism. It is only within the ED approach that new results on the correlations are obtained.

2- The paper is well written and easy to follow. The introduction gives a good overview of the field of open many body systems and the theoretical techniques to deal with them.

Weaknesses

- The problem I have with the manuscript is related to the perspective the results are put in. In the abstract and conclusions, the difference with the correlations in the ground state of the Bose-Hubbard model are emphasised. It is a bit a strange choice to put the emphasis on these differences, because the steady state is rather the infinite temperature one.

We thank the Referee for the report and the comments. We had decided to make the comparison with the ground state because in the driven dissipative case the NESS is often thought as the open-system equivalent of the ground state. In both cases, we study the excitations on top of the zero-eigenstates of the systems generator of the full quantum dynamics. We agree with the Referee that the state could be compared to an equilibrium state at infinite, or finite but large temperature. Hence, we added a new subsection on this point, highlighting the similarities and differences of our results of the open system with respect to the high-temperature one.

- Actually in Sec. 5.4 the comparison with the infinite temperature closed system is made and a good correspondence is observed. At least, this is what I get from Fig. 7 and the describing text. It is said in the abstract that "the dispersion is quadratic instead of linear at small wave vectors". But in Fig. 7, the dispersion also looks quadratic to me for the closed system.

We apologize that the text was not clear enough on this point. In the first version of the manuscript, the NESS we used to obtain the left panel of Fig. 7 is not fully equivalent to an infinite-temperature state, because the latter would have vanishing fugacity since at equilibrium at very high temperature the chemical potential is very large and negative. The NESS of our problem corresponds to an infinite temperature state with a finite fugacity. In the new version of the manuscript, we have added the comparison of our calculations with the ones of an equilibrium case at finite temperature.

I would therefore tend to say that the driving and dissipation has a small effect on the first-order correlations. I agree that this conclusion may be a bit disappointing and that the authors therefore put the emphasis on the difference with the zero-temperature state. But that conclusion would also hold for comparing the infinite and zero temperature closed systems and no one would be surprised by that.

The main message of our work is that drive and dissipation induce large fluctuations in the system. This provides an example where, in presence of drive and dissipation, it is not possible to stabilize the equivalent of a strongly correlated ground-state phase of matter, as e.g. a Mott insulator or a strongly interacting superfluid. We agree that indeed the response is much closer to the equilibrium large-temperature one, but with some differences that we have highlighted in the new version of the manuscript. For example, the excitation linewidth of the lowest branch in the spectrum depends on the loss rate and on the interaction strength; hence it is tunable, at difference from the equilibrium case.

Given the close similarities between the left and right hand panels in Fig. 7, I could imagine that it is possible to develop a perturbative treatment of the driving and dissipation, but it is hardly the task of the referee to suggest new research.

We thank the Referee for the suggestion. However, it is not so simple as it seems: a perturbative treatment in the drive and dissipation is not possible since it is precisely the compensation of drive and losses that fixes the form of the NESS. This is also evident from a field theoretical perspective, since drive and dissipation are fully included already in the quadratic theory. In essence, the effective noise induced by drive and dissipation induces fluctuations that act closely, but not exactly, as thermal fluctuations. On the other hand, we can treat the interaction strength perturbatively. In the revised version of the manuscript, we have performed a calculation to one-loop order in the interaction coupling strength.

Report

The manuscript by Zündel et al. presents a theoretical analysis of the first order correlation function of a driven-dissipative Bose-Hubbard model with incoherent pumping and losses. The current manuscript leaves me a bit wondering whether there is something interesting to be said about the first order coherence for this system with respect to the closed system. This leaves one with a nice piece of work where the conclusions seem to be unfortunately a bit trivial. But of course the authors are very much welcome to point out my oversights.

Requested changes

Unless I have missed it, the manuscript actually does not point out a big impact of the nonequilibrium condition on the first order correlation function. If the authors cannot convince me that I am wrong, I am afraid that the manuscript lacks a clear message that warrants publication in SciPost.

We thank the Referee for the constructive criticism. As already anticipated in the answer above, the first order correlation function of the driven-dissipative case is fundamentally different from the equilibrium finite-temperature one. For example, the spectral linewidth is tunable, and not fixed by the value of the temperature. Our results show for the first time how the decay rate behaves in the parameter space. In particular, we show that the decay rate depends on hopping and interaction strength, which is completely different from the case of a high-temperature equilibrium state. The plateau for the decay at large interactions obtained in the numerical simulations in extended systems is a new result, beyond weak interactions. The shape of the dispersion relation turns out to be unchanged by the interactions, which is confirmed by the one-loop perturbative calculation. All these results can be found in section 4 and 5 of the revised manuscript. In order to show the difference with respect to a high-temperature system at equilibrium, we added a new section calculating the fugacity at the given temperature, using the same average filling, and then calculate the spectral function for this density matrix under unitary evolution. We show that, in contrast to an equilibrium setting, we can change the width of the spectral lines. Finally, we would like to stress that this is the first time that the strongly interacting regime is accessed for an open Bose-Hubbard model: for the numerically accessible case of small system and low lattice filling, we consider the full dynamics, for a large range of interaction values.

As a side remark, I would like to mention that I did not appreciate very much the discussion of the work in perspective of the NISQ era in the conclusions. There does not seem to be any potential of the presented driven-dissipative Bose-Hubbard model in this respect. It is my opinion that these boiler plate texts should be avoided as much as possible in the ChatGPT era.

We thank the Referee for this remark. We have taken this sentence out and added a comment on the perspective of observation of our predictions in the currently available experimental platforms.

=====================================================
Answer to the Second Referee
=====================================================

Strengths

- Pedagogical introduction
- High clarity of discussion
- High degree of experimental relevance
- Simple and generic model which is widely studied and of interest to a broad community of researchers
- Rigorous comparison to a number of limiting cases and a direct connection to ground-state results, for which readers may be more familiar.

Weaknesses

- Direct connection to experiment could be better reflected in references.
- There is little discussion of why the Green's function is expected to have no interaction dependence.

Report

The authors consider the Bose-Hubbard model with uniform gain and loss in a Lindblad master equation. Numerical analysis of the full system is complemented by analytical results in limiting cases. After the first two introductory sections, the steady state itself is discussed and its simple nature is emphasized. The retarded Greens function are computed and compared to results for ground-states, as well as to unitary evolution from the steady state. It is demonstrated that dissipation alters the low energy excitations. It is also shown that despite the fact that the steady state has no signature of the interactions, the relaxation towards this state has a clear dependence on the interactions. The paper is highly pedagogical and easy to follow, with an intuitive and comprehensive introduction. The techniques employed are appropriate and I believe the results are of significant value to the community given current experimental interest in relatively small Bose-Hubbard arrays. Specialists will appreciate the details, while non-specialists should also be able to follow the clear exposition. The comparison between the dissipative spectral function and that of the ground-state is particularly interesting and clear, emphasising how dissipation changes the nature of excitations. This should assist the reader to make a connection to more familiar results for closed quantum systems.

Specific Comments:

Overall, I think the direct connection to experiment could be more strongly emphasized by providing a few more references. There is a sentence "While all these studies concern closed quantum systems, in several experimental situations quantum systems are subjected to external pump and/or losses, or put in contact with some type of bath. Following this, there is a list of examples of bosonic open systems, however, only a few experiments are listed. Given the preceding sentence, I was expecting a list of experiments. This is particularly true for [31]-[40]. A few more experimental references here would be appreciated. Possible examples that are not currently present are experiments with lossy Bose-Hubbard arrays in which a subsystem experiences an effective (though possible coherent) gain due to transport from other parts of the lattice. An example can be found in PRL 116, 235302 (2016) and there are many examples of non-equilibrium Bose-Hubbard experiments in the review AVS Quantum Sci. 3, 039201 (2021). These could be used to add to the broader context of this work.

We thank the Referee for this weakness of the paper pointed out and all the references provided. They were very helpful to us and we implemented them in the new version of the manuscript. Furthermore, in the introduction we have added more examples of possible platforms and future experiments.

There are other models without any interaction dependence in the steady states, but non-trivial fluctuations and decay that depend on interactions. However, to my knowledge these are critical systems with dark states, specifically, absorbing dark states where the steady state is a vacuum. For example, PRA 98, 062117 (2018) and PRL 132 120401 (2024). The manner in which the Bose-Hubbard example here is 'featureless' is of course different, being an infinite temperature steady state with a given fugacity. Presumably there are simple counter-examples where the steady state is trivial but there is no signature of interactions in the spectral function? Generically, dephasing leads to infinite temperature states. In those cases, would there be any signatures of interactions in the approach to the steady state? Since the authors emphasize that this aspect is remarkable, it would be nice to have some discussion of the broader expectations here. Tangential to this, why is it surprising that the decay rate and oscillation frequency would be renormalized by the interactions?

We thank the Referee for bringing this discussion to a broader perspective. We are not aware of counter-examples of a trivial steady state and trivial spectral function for a system being interacting in its unitary Hamiltonian part. The suggestion of studying an interacting system with dephasing is very interesting. To the best of our knowledge, there is no study on the spectral function of such a model in the open case and this could be an interesting extension of our work. We called our result remarkable since, it is a priori unexpected that drive and dissipation completely dominate and determine the steady state, but not the excitation spectrum on top of it, which has several signatures of interactions, e.g. in an excitation branch and in the linewidth. This is different from the equilibrium case, where both ground state and excited states are affected by interactions. We have added a sentence in Sec 6.1 of the revised manuscript to make this point more clear. Concerning the approach to the steady state, a complete study was not the main goal of this work, hence we just studied the behaviour of the total particle number as a benchmark. For this quantity, there is no effect of interactions, see Equation (14) in the new version of the manuscript. We expect that the first-order correlation function in the approach to the steady state will indeed depend on interactions.

Requested Changes

Fig 3a: A lot of negative times are shown in the figure. Is there a reason for this? It is a little bit distracting and some space could be saved by reducing this.

We thank the Referee for the suggestion. We have cut the figure as proposed.

Fig 3b: Are the oscillatory features best shown in the log-plot? Perhaps this could be clarified with an inset? I appreciate that the decay may make it difficult to visualise with a linear scale and the current figure may be as clear as it can be.?

We thank the Referee for pointing this out. We have made an attempt to visualize the oscillations but -- as the Referee anticipated -- the outcome was not satisfactory enough to our eyes and we have finally not added the inset.

Pending clarifications to my questions and subsequent minor changes where appropriate, I am happy to recommend this for publication in SciPost Physics.

Improved connection to experiment.

We have added new references to improve the connection to experiment.

Better discussion around why it is remarkable that the spectral function has interaction dependence, despite the absence of interaction effects in the steady state.

We thank the Referee for pointing this out and we hope that with the newly added sections the message will be more clear. In particular, the newly added 1-loop calculation brings in further insight of the small-U behaviour of the retarded Green's function. At large interactions, we find a plateau value, that is to our knowledge a new prediction for extended systems. Finally, we made the difference to the equilibrium high temperature case more clear: we showed that in the open case we can tune the width of the spectral lines, which is not the case for the infinite-temperature state.

=====================================================
Answer to the Third Referee
=====================================================

Strengths

1- The paper presents a thorough analysis of correlations in the steady state of a driven-dissipative Bose-Hubbard model, combining insights from different analytical approaches with numerics.
2- The paper shows that even though the steady state is at infinite temperature, temporal correlations reveal nontrivial dynamics

Weaknesses

1- Given that the steady state is at infinite temperature, the ground state of the closed system is not a well-motivated point of reference. In particular, the steady state is not continuously connected to the ground state in the limit $ \gamma_p,\gamma_l\rightarrow0 $ at fixed ratio $ z $.
2- The comparison to the isolated system at infinite temperature in Fig. 7 shows almost identical behavior. Therefore, it is not clear whether the "features of the spectral function are a clear signature of the non-equilibrium nature of the quantum system" as claimed in the introduction.

Report

The paper studies correlations in the steady state of a driven-dissipative Bose-Hubbard model, using a combination of analytical and numerical approaches. The presentation is clear and the results are interesting. In particular, the nontrivial dependence of correlations and spectral features on hopping and interactions, even though the system is essentially at infinite temperature, is perhaps unexpected.

We thank the Referee for the positive assessment of our paper.

However, I do not think that the claim that the "features of the spectral function are a clear signature of the non-equilibrium nature of the quantum system" is justified. On the one hand, the comparison to the isolated system at zero temperature seems a bit far-fetched; on the other hand, the comparison to the isolated system at infinite temperature shows almost identical features. Therefore, what the consequences of the system being driven and dissipative are remains somewhat unclear.

We thank the Referee for pointing out this weakness in the presentation of the results. As also answered to Referee 1, we decided to make the comparison with the ground state because in the driven-dissipative case the NESS is often thought as the open-system equivalent of the ground state. In both cases, we study the excitations on top of the zero-eigenstates of the systems generator of the full quantum dynamics. To address the Referee's criticism, in the revised version of the manuscript we added a section comparing our results to the ones obtained using a finite-temperature equilibrium states. This highlights the differences of the open system with respect to the infinite temperature one: in particular, in the open system the fugacity remains finite, hence providing a different state than the infinite-temperature one. Furthermore, we have shown that the linewidth of the excitation branches depends on the dissipative parameters. We have added a new case, shown in Figure 5, right panel, which is different from the infinite-temperature case with unitary evolution shown in Fig. 9 of the new version of the manuscript.

Moreover, at the moment I do not see how the paper satisfies the criteria of providing "a novel and synergetic link between different research areas," or of opening "a new pathway in an existing or a new research direction." Therefore, I do not think that the paper is suitable for publication in SciPost Physics, at least not in its present form.

We thank the Referee for their assessment. We hope that the revised version better matches the Referee's expectations, and we respectfully propose what in our opinion are the main synergy and novelty aspects in our manuscript. First, by making the link between open and closed interacting systems, we connect the research areas of quantum optics and of many-body physics. The chosen observable, i.e. the spectral function, is a quantity routinely studied in many-body and condensed matter physics (e.g. the outcome of ARPES measurements), but also accessible in quantum optics experiments. Another strength point of our work is the combination of a variety of methods, hence making a link between the researchers working on field-theoretical Keldysh approaches with those developing numerical methods. We also connect experts of open and closed interacting systems. Concerning the novelty aspects, we are the first to obtain the two-point space-time correlation function of the Bose Hubbard model, which is one of the most fundamental lattice models, in an open environment. We have shown that the spectral function is particularly rich in information for open quantum systems. As suggested by the Referee 2, it would be interesting to calculate it also for other types of noise, such as dephasing, and of course for larger systems, as soon as the computing power will permit it. The idea that the decay rate of excitations is fixed both by dissipative and interaction parameters is also to the best of our knowledge new. Incidentally, even the calculation of the spectral function for the finite-temperature Bose-Hubbard model at equilibrium, that we have calculated as comparison case, is completely new and was known before only in the low-temperature and infinite-interaction limit (see Ref. 18 of the manuscript).

Requested changes

We are very grateful to the Referee for the detailed remarks and questions, that we answer point by point.

Q1: In Fig. 5, on the left-hand side there seem to be $ L=8 $ different values of momenta, but on the right-hand side the values of $ k $ look continuous. Why is that the case? Further, for $ N s=3 $, there should be three white lines in the plot on the left-hand side, but I can see only two.

Both spectra of Figure 5 are discrete in the momentum space $ k $. To our eyes, the discretization is clearly visible in both images, with discontinuous jumps of the lower dispersion branch by changing from one value of $ k $ to the next. If the Referee sees it continuous, could this be a problem of the pdf reader used to display the figures? Regarding the solid white lines, we expect to have only two of them since their number is fixed by $ N_s-1 $ and in the current calculation we have taken $ N_s=3 $. Near each solid line there are two dotted lines. This is now explained in the caption of the figure.

Q2: What are the blue, red, and green lines in Fig. 6?

They refer to the exact solution at $ U=\infty $ taken from Ref. 17, and correspond to the analogue of the Lieb-I and Lieb-II branches in the uniform case, as well as the third branch expected to emerge in the lattice case. This was already said in the text, and for clarity we added this also to the caption of the figure.

Q3: What is the exact solution shown in the left panel in Fig. 6? How is $ \mu $ determined?

By 'exact solution' we meant 'exact diagonalization' solution, without truncation of the local Hilbert space. We thank the Referee for pointing out this, we have now specified it in the figure caption. The chemical potential is obtained from the ground state energy $ E_N $ of a $ N $-particle system by taking the discrete derivative $ \mu=E_{N+1}-E_{N} $, where $ E_N $ is obtained from the DMRG calculations. We added the definition and the expression for $ \mu $ in the ground state Bose-Hubbard model in the last section of the appendix.

Q4: For small system sizes as shown in Fig. 5 and the left panel of Fig. 6, the difference between a linear and a quadratic dispersion is really hard to see. Is there a better way to show this?

We agree with the Referee that for the small system sizes it is not easy to tell the difference between linear and quadratic dispersion. The best way we could think of is the comparison to tentative analytical dispersion curves: even in the small-system case, we can tell that the closed system has rather a linear dispersion and the open system rather a quadratic one. In order to show the emergent behaviour in the small system, we have also put the calculation for a large system size in the closed case, where it was possible to obtain it with the current state-of-art computational facilities. To-date, it remains a computational challenge to perform large-system calculations for the open-system case.

Q5: In the captions of some figures (e.g., Fig. 7, 8, 9, 10) parameters are not given in units of $ \gamma_l $.

We thank the Referee for spotting this. The units are now added in each caption.

Q6: Do the right panels of Figs. 5 and 7 show the same data? This is somewhat confusing.

As the Referee correctly remarked, we were displaying two times the same data, for two different purposes, i.e. to compare weak to strong interactions and unitary to dissipative dynamics. Following the Referee's remark, we now display the spectral function corresponding to other parameters values in the right panel of Figure 5. This also allows us to show how the linewidth of the lowest excitation branch changes by varying the parameters.

Q7: What is the meaning of the symbol $ \omega_{N_1,L} $ in Eq. (30)?

We thank the Referee for pointing this out and added a sentence in the beginning of the section.

Q8: The discussion in Appendix B is unclear. What is the "problem" defined by Eq. (45)? What are solutions to that problem? What determines the value of the upper bound dl? What happens going from Eq. (47) to (48)?

We thank the Referee for the questions. Our apologies, the section was indeed confusing. We reformulated the introductory problem and made explicit examples of the upper bound, which should be named $ N_s $, to make direct reference to the physical application. We changed this and added further explanation.

Q9: It is not quite clear to me why Appendices C and D are included in the manuscript.

The solutions presented in Appendices C and D are obtained with a similar method as the one used for the main text. We included it in the appendix for completeness, as we were not aware of other references containing explicitly the steady state result of this master equation, and it might be useful for further studies.

Q10: What is the parameter $ \gamma $ in Eq. (50)?

We thank the Referee for noticing this, it a typo and should be $ z $, we changed it and defined it in the corresponding section.

Q11: I believe in Eq. (52) there is a typo: it should be {1,...,L} {i}.

We thank the Referee for pointing out this typo. We corrected it.

Q12: What are the "superoperators" mentioned in Appendix E?

We thank the Referee for the clarification question. A superoperator is a concept used in open quantum systems, specifically dealing with Lindbladian dynamics and defined on the tensor product of the space of operators with itself. We added a sentence in the text to define it.

Q13: Equations (61) and (62) are supposedly expansions in $ U\gg J $, but $ J $ does not appear in these equations.

We thank the Referee for the detailed reading, this is a typo which we corrected. It is $ U\gg \gamma_l $ since $ \gamma_l $ is the scale we compare to throughout the text.

Q14: The discussion of the fitting procedure in the second paragraph of Sec. F.2 is unclear.

We thank the Referee for pointing this out, we rewrote the discussion. Further details are provided in the answer to Q19 below.

Q15: The caption of Fig. 8 refers twice to the black solid line, which I find somewhat confusing.

We thank the Referee, we have clarified the text of the caption.

Q16: I do not think that "one-sited" in Sec. F.3 is a commonly used word. What is meant here? Is this the result for a single site? But this would correspond to $ J=0 $ and not to small $ J $ as stated in the text.

We thank the Referee for the remark. We agree with the Referee that the result holds for $ J=0 $. We changed the corresponding sentence. Also, we replaced the wording "one-sited" in the text and in all figures with "single site".

Q17: What is the parity that is conserved as stated in Sec. F.5?

The parity refers to the spatial reflection symmetry in space ($ x\mapsto -x $). We change the word "parity" to "spatial parity" conservation.

Q18: Where is the approach described in Sec. F.5 used? I suppose it is used to obtain the left panel in Fig. 7, but this could also be obtained by using the same approach as for the right panel, simply by setting $ \gamma_l=\gamma_p=0 $ in the time evolution.

We thank the Referee for the remark. The approach described in Appendix F.5 is indeed used to obtain the left panel of Figure 7. As correctly pointed out, this calculation could have also been done by setting $ \gamma_l=\gamma_p=0 $ in the Lindbladian dynamics. We indeed did the calculation in both ways as a benchmark of the code. The reason we point out the approach detailed in appendix F.5 are twofold. First, for pedagogical reasons, we provide details on the numerical method of exact diagonalization. The ground state calculation was also performed using a similar but simpler fashion, as the superposition of the states required for the NESS needs not to be taken into account. This also highlights the difference between the two approaches: Only the initial state is different. Second and more importantly, we calculate the particle number conserving unitary time evolution in each particle sector. This allows to greatly reduce the numerical complexity, as the states can be efficiently stored. The results shown in figure 7 can be reproduced in this way on an ordinary laptop with 8GB RAM. The same code is used for the left panel in figure 6, which also runs on the laptop. We added a sentence in the caption of Figures 6 and 7 to point to the respective appendixes.

Q19: In Fig. 10, it seems surprising that the results for $ L=2,4 $ agree but are way off for $ L=6 $. Why is that? In the same figure, what is the meaning of the error bars?

We thank the Referee for pointing this out. The decay rate displays indeed some system size dependence, which is now shown in more detail in the newly added Figure 12. This is found in the strongly interacting limit, while no such effect is observed at weak interactions. The error bars indicated the variance of the family of curves fitted for the decay. Due to the oscillations on different scales, the result was dependent on different fit intervals chosen. Fitting a family of curves in turn can give an estimate of how much and what is the dominant error source which is captured in the fit. We rewrote the explanation in the main text, see also Request Q14.

Q20: The Heaviside function is denoted by both $ \Theta $ and $ \theta $.

We thank the Referee for noticing this inconsistency of notation. We replaced it in the respective places with $ \Theta $.

---

## Round 2 · List of Changes

1. A perturbative one-loop calculation in the interaction strength for the open quantum system has been made and the analytical result for the on-site Greens function was calculated.
2. The results have been compared to the numerical results. The results were discussed.
3. A new section explaining the expected results for finite/infinite temperature equilibrium states has been written.
4. Numerical results for the thermal system have been calculated and compared.
5. A section to explain the differences of the open system to an equilibrium system has been added.
6. Captions and display of several figures have been changed according to requests
7. Figure 5 right panel has been added, it displays the open strong interacting spectral function with a larger ratio of dissipative over coherent parameters, contrasting the difference to the quenched dynamics and the high temperature case.
8. A new figure containing the dependence of the infinite interaction decay over the system size has been added.
9. Suggestions for experimental platforms were discussed.
10. The appendices have been worked over. Especially, appendix B has been rewritten almost entirely.
11. Some minor modifications in the text have been made.

---

## Round 3 · Author Response

Dear Editors,
We warmly thank the Referees for the positive assessments. We have implemented all the minor modifications suggested by the Referee 3, highlighted in red in the resubmitted version. An answer to the report of Referee 3 is given below.
Best regards, Martina Zündel and Anna Minguzzi on behalf of the authors
-------------- Answers to the Referee 3 -------------------- We thank for the careful reading, the suggestions and the positive feedback. 1) We added a half sentences in the conclusion concerning the linear excitation branch at low temperatures in equilibrium. 2) We tempered the expression from "remarkable" to "at first glance this seems astonishing" and we point out what is the difference to the equilibrium case: In the equilibrium density matrix the interaction does appear explicitly, compared to the open system steady state. 3) We thank for the question regarding the one-loop correction to the oscillation frequency: We could not make a quantitative comparison as it is hard to extract the correct oscillation frequency from the data with accuracy.

---

## Round 3 · List of Changes

- added a half-sentence in the conclusions
- changed a sentences in section 6.1
- corrected an expression in the headline of fig. 9

---

## Editorial Decision

editorial_decision: